# Tiltable objective microscope visualizes selectivity for head motion direction and dynamics in zebrafish vestibular system

Masashi Tanimoto [1,2] ✉, Ikuko Watakabe[1,2] & Shin-ichi Higashijima [1,2] ✉

Spatio-temporal information about head orientation and movement is fundamental to the sense of balance and motion. Hair cells (HCs) in otolith organs of the vestibular system transduce linear acceleration, including head tilt and vibration. Here, we build a tiltable objective microscope in which an objective lens and specimen tilt together. With in vivo $Ca^{2+}$ imaging of all utricular HCs and ganglion neurons during 360° static tilt and vibration in pitch and roll axes, we reveal the direction- and static/dynamic stimulus-selective topographic responses in larval zebrafish. We find that head vibration is preferentially received by striolar HCs, whereas static tilt is preferentially transduced by extrastriolar HCs. Spatially ordered direction preference in HCs is consistent with hair-bundle polarity and is preserved in ganglion neurons through topographic innervation. Together, these results demonstrate topographically organized selectivity for direction and dynamics of head orientation/movement in the vestibular periphery.

Our sense of the world relies on spatial and temporal information processing in the nervous system. Since sensory inputs change frequently over time due to the movement of surrounding objects and the self, the temporal processing of sensory cue modulation contributes to the analysis of ongoing events[1,2]. The vestibular system encodes both spatial information (head movement in space and orientation relative to gravity) and temporal information regarding the modulation of inputs (e.g., fast/slow head movement)—these are transduced from head movement in the vestibular periphery. Linear acceleration and gravitational acceleration are sensed by otolith (macular) organs whereas rotational acceleration is sensed by semicircular canals. In both organs, hair cells (HCs) transduce the acceleration into electrical signals by mechano-electrical transduction at the hair bundles.

As for spatial signals, hair-bundle polarity determines HC selectivity to the direction of acceleration[3,4]. Otolith organs contain HCs with spatially ordered hair-bundle polarity that reverses at an imaginary line of polarity reversal (LPR)[5]. Thus, it is believed that, depending on the hair-bundle polarity, different groups of HCs detect different directions of acceleration, and send the signals to the brain via

vestibular ganglion neurons (VGNs). However, the direction-selective HC responses to natural head movement have never been systematically quantified in vivo, because it is difficult to measure the HC activity during head motion. Moreover, it remains unclear whether the responsiveness of HCs with a similar hair-bundle polarity differs across different regions. Furthermore, although the sites of the VGN peripheral innervation of HCs determine afferents' direction preference[6,7], how the direction information is represented in the VGN population has not yet been mapped.

The otolith system also receives static/dynamic linear acceleration (e.g., slow static tilt and rapid vibratory translation) that are thought to be relayed by different classes of VGNs. The otolith organs contain a morphologically specialized region, referred to as the striola. Striolar HCs and VGNs innervating the striolar HCs are morphofunctionally distinct from their extrastriolar counterparts[8–10]. Electrophysiological recordings from the VGNs indicate that VGNs innervating the striolar HCs relay dynamic/high-pass/phasic signals such as head vibration and jerk whereas those innervating extrastriolar HCs convey static/low-pass/sustained signals, including static head tilts[7,11]. However, whether the decomposition of the static/dynamic

---

[1]Division of Behavioral Neurobiology, National Institute for Basic Biology, Okazaki, Aichi 444-8787, Japan. [2]Neuronal Networks Research Group, Exploratory Research Center on Life and Living Systems (ExCELLS), Okazaki, Aichi 444-8787, Japan. ✉e-mail: tanimoto@nibb.ac.jp; shigashi@nibb.ac.jp

linear acceleration originates at the VGNs or at upstream HCs remains unclear.

To answer the above-mentioned questions, the systematic measurement of the neural activity in vivo during physiological head movement is needed. Larval zebrafish are an exceptional model system because the high transparency and small size of organs enable in vivo whole-organ imaging. The vestibular system is largely conserved among vertebrates, although there exist some differences such as the presence of a lagena[8,12,13]. In zebrafish, macular HCs become functional as early as 1 day postfertilization (dpf)[14,15]. The vestibuloocular reflex starts as early as 3 to 4 dpf[16,17] and, around this age, larvae start keeping a dorsal-up posture[18]. These behaviors solely rely on utricular inputs in the larvae[17,18]; the developing semicircular canals are immature at the larval stage[19]. The transparent brain enables brain-wide functional imaging during vestibular stimulation[20,21]. Inferred VGN responses to phasic stimulus and morphological connectivity from HCs to the brain have been reported[22,23]. However, due to a lack of appropriate methods, how the spatio-temporal information on head movement is transduced in the vestibular periphery has not yet been functionally examined.

Here we design and build a tiltable objective microscope in which an objective lens and a larval zebrafish tilt together 360° during imaging. Combined with a new in vivo preparation for imaging the vestibular periphery, Ca$^{2+}$ imaging of all utricular HCs and VGNs during static tilt and vibration in the pitch and roll axes reveals direction- and static/dynamic stimulus-selective topographic responses. Quantitative analysis reveals that the static tilt is preferentially, though not exclusively, transduced in the extrastriolar HCs whereas the vibration is preferentially received by the striolar HCs. Thus, decomposition of static and dynamic head motion originates at the HCs. The tilt direction signals are topographically represented in the VGNs via the spatially ordered innervation of HCs. Together, our microscope design allowed to uncover the direction- and static/dynamic stimulus-selective topographic organization in the vestibular periphery.

## Results

### Tiltable objective microscope enables functional imaging during 360° static tilt and vibration stimulus

We designed and built a custom microscope with commercially available optical components (Fig. 1a). An objective lens unit, which contained the specimen, was combined with a motorized rotation stage that had a central aperture. Slow, large-angle rotation of the stage (stimulus 1) tilted the specimen whereas bidirectional, fast, and small-angle rotation (stimulus 2) vibrated the specimen (Fig. 1b and Supplementary Movie 1). Because the specimen was off the axis of stage rotation, any stage rotation produced both linear and angular acceleration. During stimulus 1, vector components of the gravitational acceleration on the specimen changed depending on the rotation angle whereas the stage movement produced small inertial acceleration (Supplementary Fig. 1a–c). The amplitude of the inertial acceleration was small over the frequency range up to hundreds of Hz (Supplementary Fig. 1e, f). Thus, stimulus 1 produced static tilt of the specimen. During stimulus 2, in contrast, back-and-forth motion produced vibratory inertial acceleration only in the single axis (the tangential direction to the rotation circle) with small changes in the vector components of the gravitational acceleration (Supplementary Fig. 1d). The vibratory inertial acceleration contained large amplitudes of frequency peaks ranging from 2.2 to ~100 Hz (Supplementary Fig. 1g), which corresponded to frequency range of head yawing movement during swimming in larval zebrafish[24]. Thus, stimulus 2 vibrated the specimen in a nearly linear axis with the multi-band frequency in the physiological range.

Images were recorded during the stage rotation. The images rotated, since the camera did not move while the stage rotated. Thus, the images were registered after experiments (Fig. 1c and

Supplementary Movie 2, see "Methods"). During the rotation, artificial fluorescent intensity changes were produced due to unwanted changes in optics (Fig. 1e, see Discussion). To reduce the artifacts, we adopted a two-color ratiometric imaging method[25,26]. Green and red fluorescence were separated and simultaneously recorded by a single digital camera (Fig. 1a). The images were merged and the neuronal responses were quantified by the green-to-red fluorescence intensity ratio (Fig. 1c).

We characterized the performance of the microscope by imaging fluorescent beads (Fig. 1d). The image rotation angle was calculated from the beads position (Fig. 1e, bottom, Supplementary Movie 2). Using the rotation angle, the images were registered by counter-rotating the images (Fig. 1c and Supplementary Movie 2). In the registered images, time-series fluorescence intensity was measured per bead. Due to optical distortion, the green and red signals fluctuated during the tilt (Fig. 1e and Supplementary Fig. 1h). The maximum intensity change exceeded 50%. These artifacts were significantly reduced in the ratio of the two signals. The ratio change, $\Delta R/R_0$, was confined within ±10% during 360° tilt (Fig. 1e–i).

Next, we imaged fish brain neurons. In 5 dpf transgenic larvae, *Tg(nefma-hs:Gal4; UAS:Kaede)*, a photoconvertible green fluorescent protein, Kaede, was expressed in a subset of the brainstem neurons[22]. The green-Kaede was partially photoconverted to red-Kaede by ultraviolet light such that neurons contained both green- and red-Kaede. The neurons were imaged during head tilt (Fig. 1j). The fluorescence signals in each neuron fluctuated during the tilt (Fig. 1k and Supplementary Fig. 1i). The maximum intensity change exceeded 80%. These artifacts were significantly reduced in the ratio of the two signals. $\Delta R/R_0$ was confined within ±10% during the 360° and 90° tilts, and within ±5% during the vibration (Fig. 1k–o and Supplementary Fig. 1j–m). The amplitude of these artifacts was considerably lower than that of neural responses (see the following results). Thus, these data suggested that the tiltable objective microscope enables the functional imaging of neural activity during 360° static tilt and vibration stimulus.

### Utricular HCs exhibit topographic, direction-selective responses to static tilt

Most utricular HCs lie in the horizontal plane and are sensitive to pitch and roll tilts. Previous morphological studies showed the hair-bundle polarity in larval zebrafish (Fig. 2a)[23,27,28], but HC activity during head tilt has never been studied. In order to image utricular HC activity in vivo, we generated a transgenic zebrafish, *Tg(myo6b:jGCaMP7f)*, that expressed a genetically encoded Ca$^{2+}$ indicator (GECI) in the HCs, and crossed the fish with *Tg(myo6b:tdTomato)*[29], which expressed a red fluorescent protein in the HCs. Tissues ventral to the ears were removed, and the larva was placed dorsal-up in the tiltable objective microscope. All utricular HCs were imaged from the ventral side during static tilt in the pitch or roll axis. The static tilt produced changes in the vector components of gravitational acceleration in the specimen (Supplementary Fig. 1a–c). Because of this, otoliths would slide down relative to HCs, which provides a sustained deflection of hair bundles and stimulates HCs (Supplementary Fig. 1a). After the recording, the fish yaw orientation was rotated 90° by the rotation mount in order to image responses to the orthogonal tilt axis. This setup enabled whole-organ in vivo Ca$^{2+}$ imaging of all utricular HC responses to physiological head tilts in pitch and roll axes. Image analysis showed that, during 90° static tilt in the pitch axis, nose-down tilt evoked HC activity in the rostral utricle except the rostral-lateral edge whereas tail-down tilt activated HCs in the rostral edge and in the caudal utricle except the lateral edge; during the tilt in the roll axis, lateral-down tilt activated most HCs except the lateral edge whereas medial-down tilt activated HCs in the lateral edge (Fig. 2b, Supplementary Fig. 2a, and Supplementary Movie 3). The tilt direction–selective responses were visualized by the difference between the tail-down and nose-down

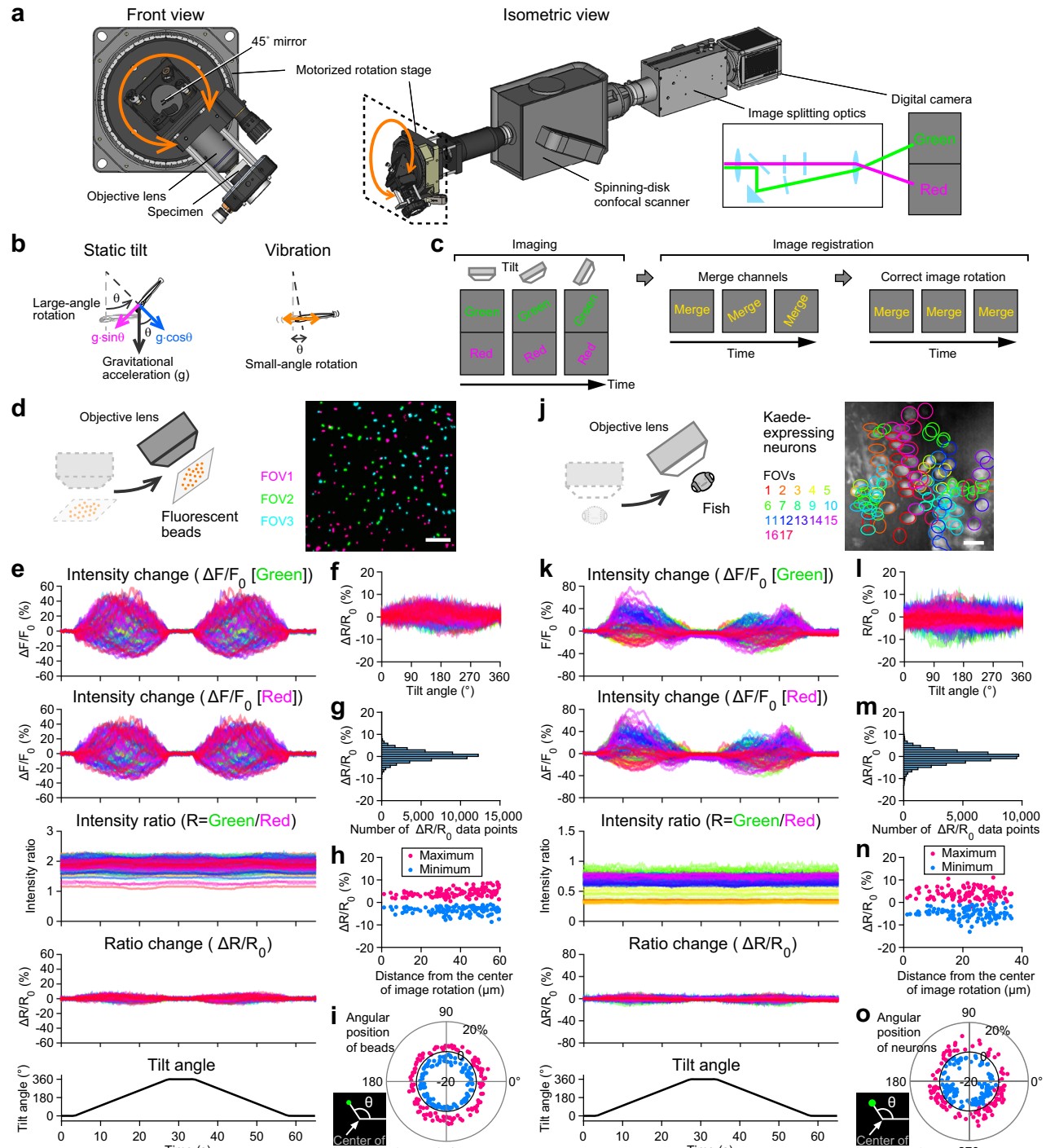

**Fig. 1 | Tiltable objective microscope enables functional imaging during 360° static tilt and vibration. a** Schematic showing a tiltable objective microscope. An objective lens and a specimen are tilted by a motorized rotation stage (see Supplementary Movie 1). **b** Gravitational vector components (magenta and blue arrows) during static tilt produced by large-angle, slow rotation (left). Inertial acceleration during vibration (orange arrows) produced by small-angle, fast rotation (right) (see Supplementary Fig. 1). **c** Rotated images are merged and registered by correcting the image rotation (see Supplementary Movie 2 and "Methods"). **d** Fluorescent beads. Images from 3 fields of view (FOVs) are superimposed. Scale bar: 20 μm. **e** Time course of beads fluorescent intensity changes during 360° tilt. Large fluctuation in green and red signals is reduced in ratio and ratio change ($\Delta R/R_0$). 116 beads from 3 FOVs. **f** Relationship between tilt angle and $\Delta R/R_0$. **g** Distribution of $\Delta R/R_0$ values during tilt. **h** Relationship between beads distance from the center of image rotation, which is the center of the image in [**d**], and $\Delta R/R_0$

maximum (red) and minimum (blue). **i** Relationship between the angular position of beads around the center of image rotation, which is the center of the image in [**d**], and $\Delta R/R_0$ maximum (red) and minimum (blue). **j** Brain neurons expressing green- and red-Kaede in *Tg(nefma-hs:Gal4; UAS:Kaede)* larval zebrafish at 5 dpf. Images from 17 FOVs are superimposed. Red channel is shown in grayscale. Positions of neurons are labeled by FOV colors. Scale bar: 10 μm. **k** Time course of neuronal fluorescent intensity changes during 360° tilt. Large fluctuation in green and red signals is reduced in ratio and $\Delta R/R_0$. 116 neurons from 17 FOVs. **l** Relationship between tilt angle and $\Delta R/R_0$. **m** Distribution of $\Delta R/R_0$ values during tilt. **n** Relationship between neuron distance from the center of image rotation, which is the center of the image in [**j**], and $\Delta R/R_0$ maximum (red) and minimum (blue). **i** Relationship between angular position of neurons around the center of image rotation, which is the center of the image in [**j**], and $\Delta R/R_0$ maximum (red) and minimum (blue). Source data are provided as a Source Data file.

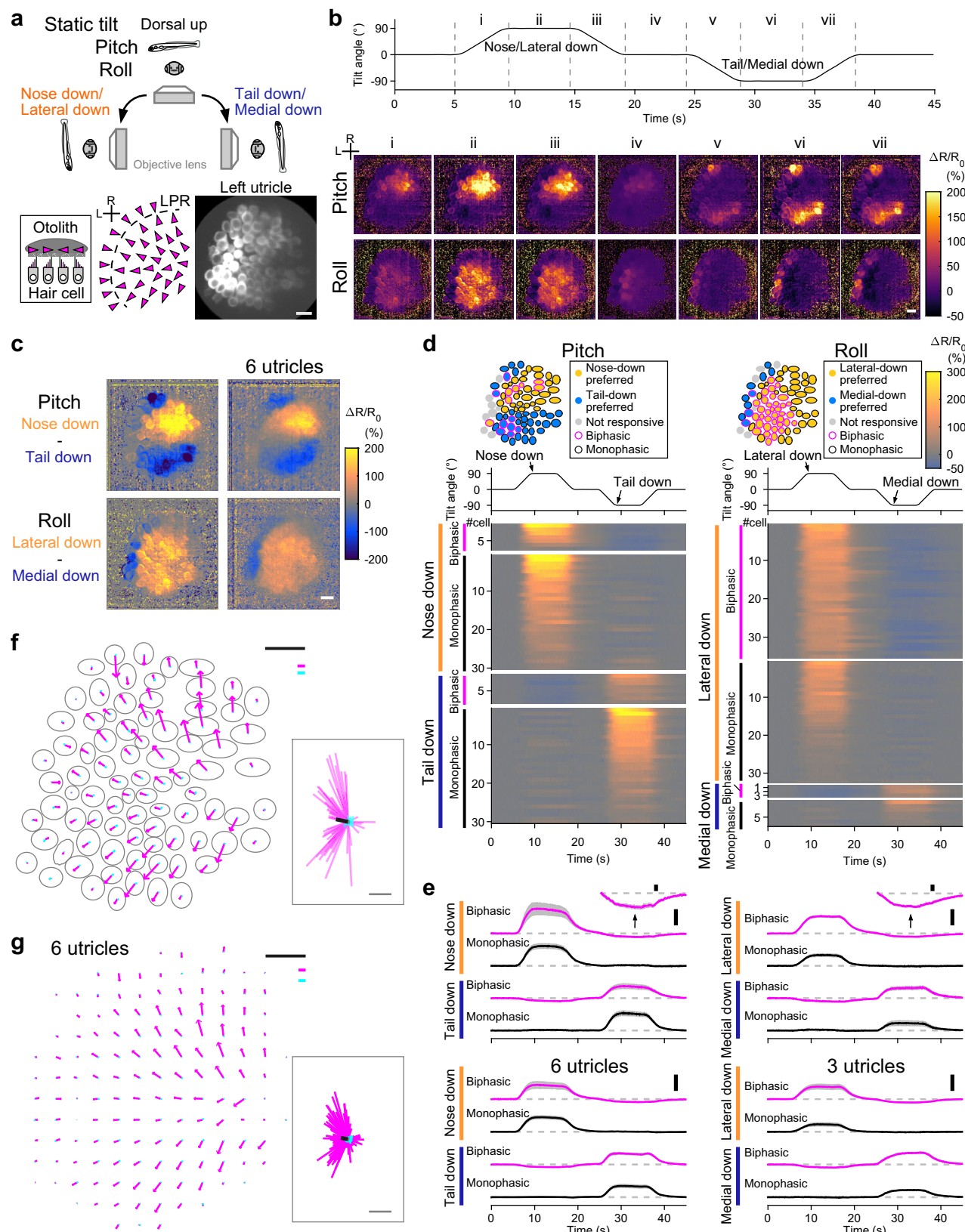

responses, and between the medial-down and lateral-down responses (Fig. 2c). This activity pattern was consistent with the morphological hair-bundle polarity (6 utricles, Fig. 2c)[23,27,28].

To analyze the activity in each HC, $\Delta R/R_0$ was quantified. HCs in which $\Delta R/R_0$ exceeded 10% were regarded as "activated," based on the amplitude of the artificial $\Delta R/R_0$ during tilts (Supplementary Fig. 1j, k).

In a utricle in Fig. 2d, in response to the pitch tilt, individual HCs were activated by the nose-down or tail-down tilt. Based on the tilt direction that activated the HCs (preferred tilt direction), the HCs were categorized as the nose-down preferred or tail-down preferred group. Similarly, in response to the roll tilt, individual HCs were activated by the lateral-down or medial-down tilt; HCs were categorized as the

**Fig. 2 | In vivo Ca²⁺ imaging visualizes direction-selective utricular HC responses to static tilt. a** Imaging of utricular HCs during static tilt in pitch/roll axis (top). Side view of otolith organ (bottom inset). Top-down view of hair-bundle polarity (magenta arrowheads) reversing at a line of polarity reversal (LPR, dashed line) (bottom left). R: rostral. L: lateral. The same orientation applies to (**b**–**d**), (**f**), and (**g**). Left utricular HCs in a 5 dpf *Tg(myo6b:jGCaMP7f; myo6b:tdTomato)* larva (bottom right). Green channel is shown. Scale bar: 10 μm. **b** HC responses to 90° static tilt (see Supplementary Movie 3). Tilt time course (top). Sequential images showing mean $\Delta R/R_0$ responses during the time period "i" to "vii" (bottom). Average of 5 trials. Scale bar: 10 μm. Data from the same utricle is shown in (**c**), (**d**), (**e**), and (**f**). **c** Response selectivity for tilt direction. Tail-down response ("vi" in [**b**]) subtracted from nose-down response ("ii" in [**b**]) (top). Medial-down response ("vi" in [**b**]) subtracted from lateral-down response ("ii" in [**b**]) (bottom). The same utricle shown in (**b**) (left). Average of 6 utricles (right). Scale bar: 10 μm. **d** Tilt responses to pitch (left) and roll (right) tilts per HC. HC location (top). Tilt time course (middle).

Response ($\Delta R/R_0$) time course grouped by preferred tilt direction and response directionality (bottom). Average of 5 trials. Data from the same utricle shown in (**b**). **e** Mean $\Delta R/R_0$ time course across HCs in each preference group in (**d**) (top). Negative responses are vertically magnified (insets, scale bar: 10% $\Delta R/R_0$). Number of cells is reported in the Supplementary Fig. 2 legend. Mean responses (6 utricles for pitch, 3 left utricles for roll [3 right utricles are shown in Supplementary Fig. 2f]) (bottom). Deviation (SEM) is shown in gray. Tilt time course is the same as in (**d**). Scale bar: 100% $\Delta R/R_0$. (**f**, **g**) Example (**f**) and summary (**g**) of HC response vectors. Orientation of arrows indicates preferred (magenta)/anti-preferred (cyan) direction. Length of arrows indicates $\Delta R/R_0$ amplitude (**f**) and mean amplitude per grid (**g**, see "Methods"). Gray: HC contour (**f**). Magenta and cyan scale bars: 100% $\Delta R/R_0$. Black scale bar: 10 μm. (inset) All response vectors aligned together. Mean vectors to the preferred (black)/anti-preferred (gray) direction are overlaid. Scale bar: 100% $\Delta R/R_0$. Source data are provided as a Source Data file.

lateral-down preferred or medial-down preferred group. Within each group, a subset of HCs exhibited negative $\Delta R/R_0$ during the tilt in the opposite direction to the preferred tilt direction. When a hair bundle is deflected toward shorter cilia, mechanically gated channels, some of which are open at the resting state, are closed, cation influx including Ca²⁺ reduces, and HCs are hyperpolarized[4]. Therefore, the negative $\Delta R/R_0$ indicated hyperpolarization by the tilt in the anti-preferred direction. This functional group was categorized as biphasic HCs, and those lacking the negative responses were classified as monophasic HCs. The functional groups were reproducibly observed and also observed in the responses to 360° tilt (Fig. 2e and Supplementary Fig. 2b–f). The mean maximum amplitude in the functional groups across 6 utricles is listed in Supplementary Table 1.

Spatial distribution of the responses was visualized by response vectors in a horizontal plane (Fig. 2f). A response vector of each HC was produced by the vector sum of the pitch and roll Ca²⁺ responses. For instance, a HC that was activated by the nose-down and lateral-down tilts and suppressed by the tail-down and medial-down tilts had a response vector for the preferred direction pointing toward the rostrolateral direction and a response vector for the anti-preferred direction pointing toward the caudomedial direction. The direction and length of the vector represented the preferred/anti-preferred direction and response amplitude, respectively. In a utricle in Fig. 2f, response vector orientation was consistent with the morphological hair-bundle polarity[23,27,28]. Response amplitude appeared, on the other hand, inhomogeneous across the utricle; the amplitude was larger in the mid-rostral and mid-caudal regions in the medial half in the utricle. To summarize data from different samples, utricle positions were aligned to each other, and an imaginary grid was prepared on the utricles (see "Methods"). Each HC position was assigned to the nearest crossing point in the grid, and the mean response vector was calculated for every crossing point. The spatially ordered activity pattern was reproducibly observed in the 6 utricles examined (Fig. 2g). Thus, in vivo Ca²⁺ imaging with the tiltable objective microscope visualized the topographic pattern of direction selectivity and different responsiveness to the static head tilt in the utricular HCs for the first time.

### A subpopulation of utricular HCs receive vibratory head movement

Macular HCs also receive vibratory head motion. HC vibration responses over the macula were only inferred from afferent activity and have not yet been systematically examined. Utricular HC activity was imaged during the vibration stimulus (Fig. 3a). The vibration produced oscillatory inertial acceleration in the specimen (Supplementary Fig. 1d, g). Otoliths are much denser than surrounding tissues, and therefore move less and lag behind HCs. This relative displacement between the otolith and HCs would deflect hair bundles and stimulate HCs (Supplementary Fig. 1a). During the stimulus in the pitch (rostral-caudal) axis, a swath of HCs spanning from the rostral to the

caudolateral region gradually increased $\Delta R/R_0$ (Fig. 3b and Supplementary Movie 4); during the vibration in the roll (medial-lateral) axis, HCs in the lateral region increased $\Delta R/R_0$ (6 utricles).

To analyze activity in each HC, $\Delta R/R_0$ was measured. HCs in which $\Delta R/R_0$ exceeded 5% were regarded as "activated," based on the artificial $\Delta R/R_0$ during vibration (Supplementary Fig. 1l, m). In a utricle, a subset of the utricular HCs were activated by the pitch or roll vibration with varying response amplitude (Fig. 3c). During the pitch vibration, the rostral and caudolateral HCs except the lateral edge were activated whereas during the roll vibration, the lateral HCs except the lateral edge were activated (6 utricles, Fig. 3d). The response amplitude appeared smaller than that of static tilt-evoked responses (Fig. 2e). Presumably, this was in part because the acceleration amplitude during the vibration stimulus was smaller than the static tilt (Supplementary Fig. 1b, d). We compared the maximum HC response amplitude per maximum acceleration in the static tilt and vibration paradigms. The value for the vibration was smaller than that for the static tilt (tilt: 79.7 ± 1.1 %/g in pitch responses; 69.6 ± 6.1 %/g in roll responses; vibration: 24.2 ± 2.6 %/g in pitch responses; 27.0 ± 1.1 %/g in roll responses, mean ± standard error of the mean [SEM], 6 utricles). This suggests that even if the vibration and static tilt with the same amplitude of acceleration were applied, the hair-bundle displacement during the vibration stimulus would be smaller than that during the static tilt stimulus.

The spatial distribution of the responses was visualized in a horizontal plane. In Fig. 3e, most HCs in the lateral utricle except the peripheral edge were activated by both pitch and roll vibrations (magenta). The rostromedial HCs responded to the pitch vibration only (green) whereas middle-lateral HCs were activated by the roll vibration only (blue). This spatial distribution of the HC direction selectivity was consistent with the morphofunctional hair-bundle polarity (Figs. 2g and 3f)[23,27,28]. In the 6 utricles, pitch vibration activated 34 ± 5.9 HCs (46.4 ± 5.8%) whereas 51 ± 8.8 cells were not activated; roll vibration activated 39 ± 3.9 HCs (41.3 ± 8.3%) whereas 47 ± 7.1 cells were not activated (mean ± SEM). Thus, in vivo Ca²⁺ imaging visualized a subpopulation of utricular HCs in a spatially restricted region receiving vibratory head motion.

### Vibration and static head tilt are preferentially transduced by striolar and extrastriolar HCs, respectively

The phasic vestibular stimulus, including vibratory head movement, is thought to be relayed by the afferents innervating the striolar HCs[7]. Although this infers that the striolar HCs transduce the head vibration, the striolar HC responses have not yet been directly examined. The vibration-induced HC activity in the spatially restricted region on the utricle (Fig. 3) indicated that the activated HCs were the striolar HCs. In order to label the striola in intact fish, we focused on a Ca²⁺-binding protein S100 that is enriched in the striola[30]. Using a knock-in genome-engineering method[31], we generated a transgenic fish, *Tg(s100s-*

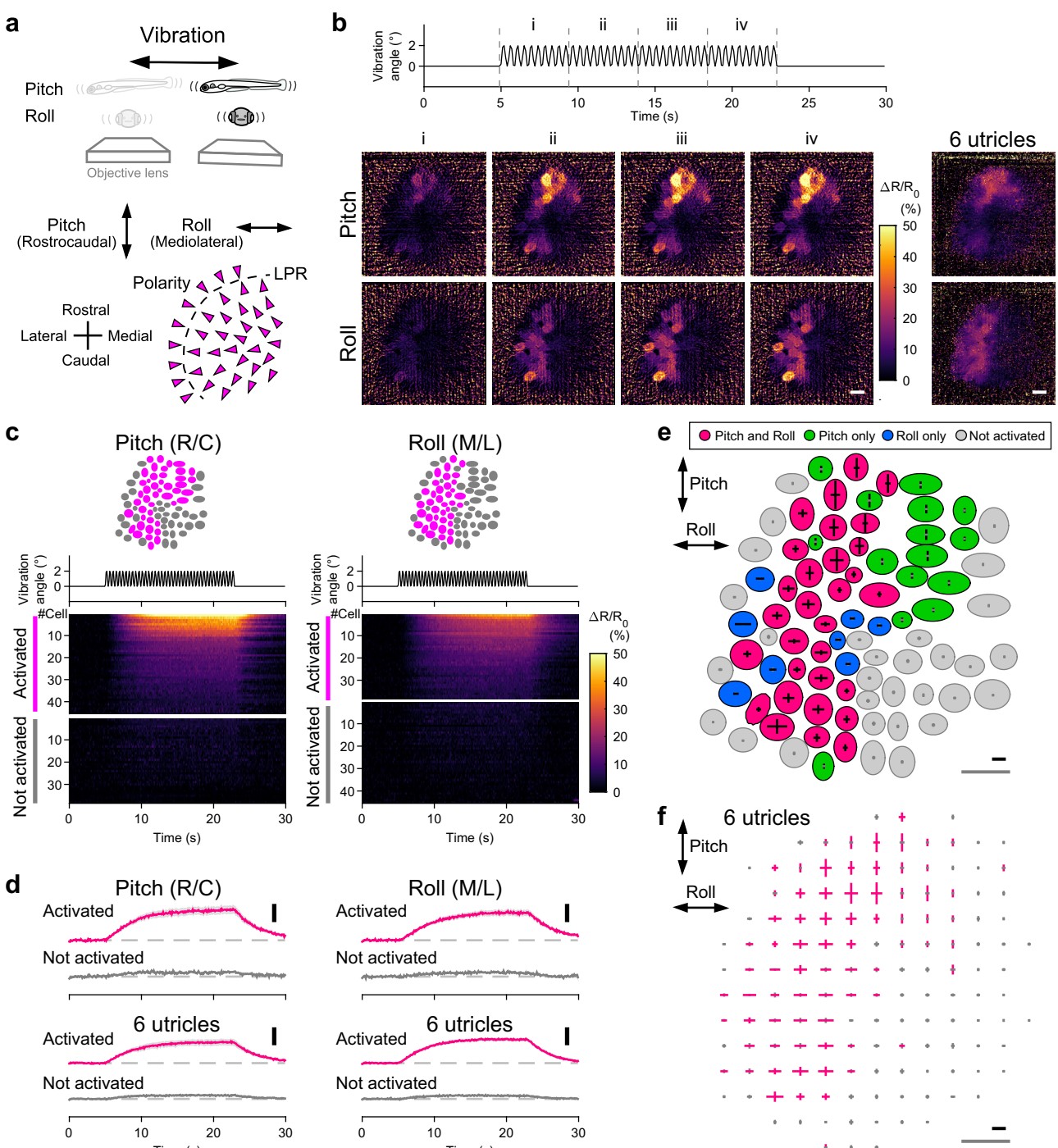

**Fig. 3 | In vivo Ca²⁺ imaging visualizes a subset of utricular HCs receiving head vibration. a** Schematic showing imaging of utricular HCs during vibration in pitch/roll axis (top). Hair-bundle polarity (arrowheads) and vibration axes (double-headed arrows) (bottom). The same orientation applies to (**b**), (**c**), (**e**), and (**f**). **b** Example of HC responses to vibration stimulus in pitch and roll axes (see Supplementary Movie 4). Vibration time course (top). Sequential images showing mean $\Delta R/R_0$ responses during the time period "i" to "iv" (left). Average of 8 trials. Organ-average of the mean $\Delta R/R_0$ responses during the time period "iv" in 6 utricles (right). Scale bar: 10 µm. **c** Example of vibration responses per HC. The same utricle shown in (**b**). Responses to pitch (left) and roll (right) vibration. HC location (top).

Vibration time course (middle). $\Delta R/R_0$ per HC (bottom). R/C: rostrocaudal; M/L: mediolateral. **d** Mean $\Delta R/R_0$ time course in HC groups in (**c**) (top). Pitch: 45 cells activated; 38 cells not activated. Roll: 38 cells activated; 45 cells not activated. Mean of the 6 utricles (bottom). Deviation (SEM) is shown in gray. Vibration time course is the same as in C. Scale bar: 10% $\Delta R/R_0$. R/C: rostrocaudal; M/L: mediolateral. Example (**e**: the same utricle shown in [**b**]) and summary (**f**) of HC response vectors. Orientation of bars indicates stimulus direction. Length of bars indicates $\Delta R/R_0$ amplitude (**e**) or mean amplitude per grid that is larger (magenta) or not larger (gray) than 5% (**f**, see "Methods"). Black scale bar: 20% $\Delta R/R_0$. Gray scale bar: 10 µm. Source data are provided as a Source Data file.

*hs:tdTomato*), and crossed the fish with *Tg(myo6b:jGCaMP7f)*. In a horizontal optical slice of a utricle, tdTomato was abundantly expressed in the region ranging from the rostral to the caudolateral utricle (Fig. 4a, b). In this region, HCs and supporting cells were

strongly labeled compared to those in the outside region. Since this S100-enriched region contained the striolar HCs that have been characterized by soma shape and hair-bundle length[23], this region was used as a proxy for the striola in this study (see "Discussion").

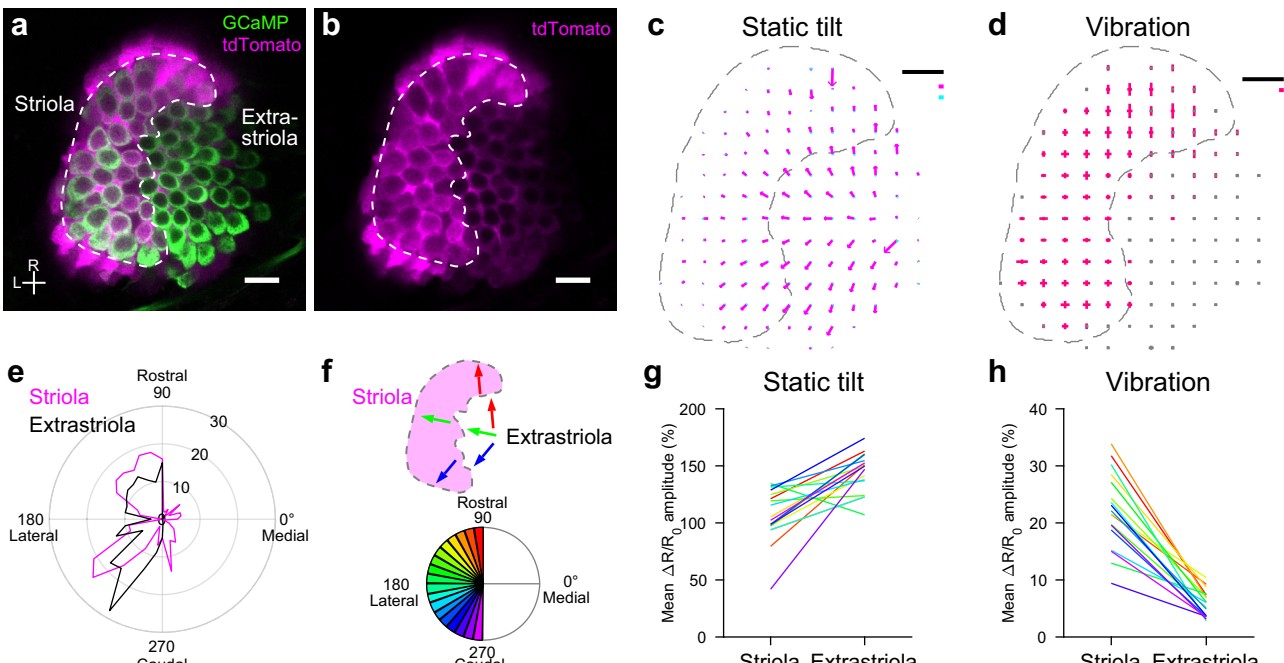

**Fig. 4 | Head vibration and static tilt are preferentially transduced by striolar and extrastriolar HCs, respectively. a, b** A horizontal optical slice of utricle in a 5 dpf transgenic fish, *Tg(myo6b:jGCaMP7f; s100s-hs:tdTomato)*. Striola is enclosed by a dashed line. Scale bar: 10 μm. R: rostral. L: lateral. Similar results were obtained in 6 experiments. **c, d** Summary of HC response vectors from 6 utricles. Mean $\Delta R/R_0$ vectors per grid. Striola is enclosed by a dashed line. Black scale bar: 10 μm. **c** Orientation and length of arrows indicate preferred (magenta)/anti-preferred (cyan) direction and $\Delta R/R_0$ amplitude, respectively. Magenta and cyan scale bars: 100% $\Delta R/R_0$ (**d**) orientation and length of bars indicate stimulus direction and $\Delta R/R_0$ amplitude, respectively. Magenta scale bar: 20% $\Delta R/R_0$. **e** Polar histogram showing number of HCs (radial axis) against HC polarity (angular axis) determined by orientation of tilt response vectors. Bin size: 10°. **f** Example of HC groups, each of which has a similar polarity (top). Color code for HC polarity groups with bin size 10° (bottom). **g, h** Pairwise comparison of mean $\Delta R/R_0$ response vector length between striolar and extrastriolar HCs in 6 utricles (18 HC polarity groups). Color code shown in (**f**). Two-sided Wilcoxon signed-rank test (**g**, $p = 1.1 \times$ e−4; **h**, $p = 7.6 \times$ e−6). Source data are provided as a Source Data file.

To examine the functional differences between the striolar and extrastriolar HCs, activity was imaged during 90° static tilt and vibration stimulus. The spatial distribution of response vectors was similar to those shown in Figs. 2 and 3 (Fig. 4c, d and Supplementary Fig. 3a). To compare the responsiveness, the vector lengths of all tilt and vibration responses were analyzed. Compared to the extrastriolar HCs, the striolar HCs had lower tilt responsivity whereas they had larger vibration responsivity (Supplementary Fig. 3b, c).

Furthermore, groups of striolar/extrastriolar HCs that had a similar direction preference (i.e., hair-bundle polarity) were compared pairwise (Fig. 4e, f). Since the medial-down preferred HCs existed in the striola but not in the extrastriola, these HCs were excluded from the analysis (Fig. 4e). The tilt response vector length in the striolar HCs was shorter than that of the extrastriolar HCs (Fig. 4g). In contrast, the vibration response vector length in the striolar HCs was longer than that of the extrastriolar HCs (Fig. 4h). Together with the results in Figs. 2 and 3, these data demonstrated that head vibration is preferentially transduced by striolar HCs whereas static tilt is preferentially, though not exclusively, received by extrastriolar HCs.

## VGNs relay head tilt signals through a spatially ordered pathway

How do the vestibular signals travel from the macular HCs to the brain through VGNs? The rostral division of the VG contains neurons, each of which innervates the utricle, the anterior part of the saccule, or the anterior or horizontal semicircular canal[32]. We found that the transgenic fish, *Tg(myo6b:jGCaMP7f; myo6b:tdTomato)*, also expresses the GCaMP and tdTomato in the VGNs. VGN activity was imaged at 6 optical slices, and the responses were visualized in the maximum intensity projection images (Fig. 5a–c) or in each optical slice (Fig. 5d). During the tilt in the pitch and roll axes, VGN activity increased in a region spanning from the center to the lateral VG (Fig. 5b and Supplementary Movie 5). During the pitch tilt, the rostral VGN subset in this region was activated by the nose-down tilt whereas the caudal subset was activated by the tail-down tilt. During the roll tilt, a VGN subset in this region was activated by the lateral-down tilt but rarely activated by the medial-down tilt. The spatial distribution of the responses was visualized by the difference between the tail-down and nose-down responses, and between the lateral-down and medial-down responses (Fig. 5c). The response map showed a topographic pattern of the tilt direction–selective responses. This pattern existed in all optical slices (Fig. 5d).

To analyze activity in each VGN, $\Delta R/R_0$ was quantified. In the VG shown in Fig. 5e, approximately 40% of imaged VGNs responded to either the nose-down or tail-down tilt (left column). In the center-to-lateral region, the nose-down preferred VGN group and tail-down preferred VGN group were localized in the rostral and caudal patches, respectively (top left). In response to the roll tilt, approximately one-quarter of the VGNs were activated by the lateral-down tilt (right column); this group was localized in the lateral region (top right).

The mean $\Delta R/R_0$ time course of the VGN groups is shown in Fig. 5f. All VGN groups had both transient and sustained components. The $\Delta R/R_0$ increased while the head tilted to the preferred direction, and the amplitude reached the maximum around the time when the tilt angle reached 90°. The amplitude decayed slowly to half while the tilt angle was kept constant, and it returned to the baseline as the tilt returned to the original angle (Fig. 5f). The functional VGN groups were reproducibly observed in 6 ganglia and were also observed during 360° tilt (Supplementary Fig. 4a–e).

In all slices in VG, the response vector orientation was spatially ordered; in the lateral region, the rostrally located VGNs had response

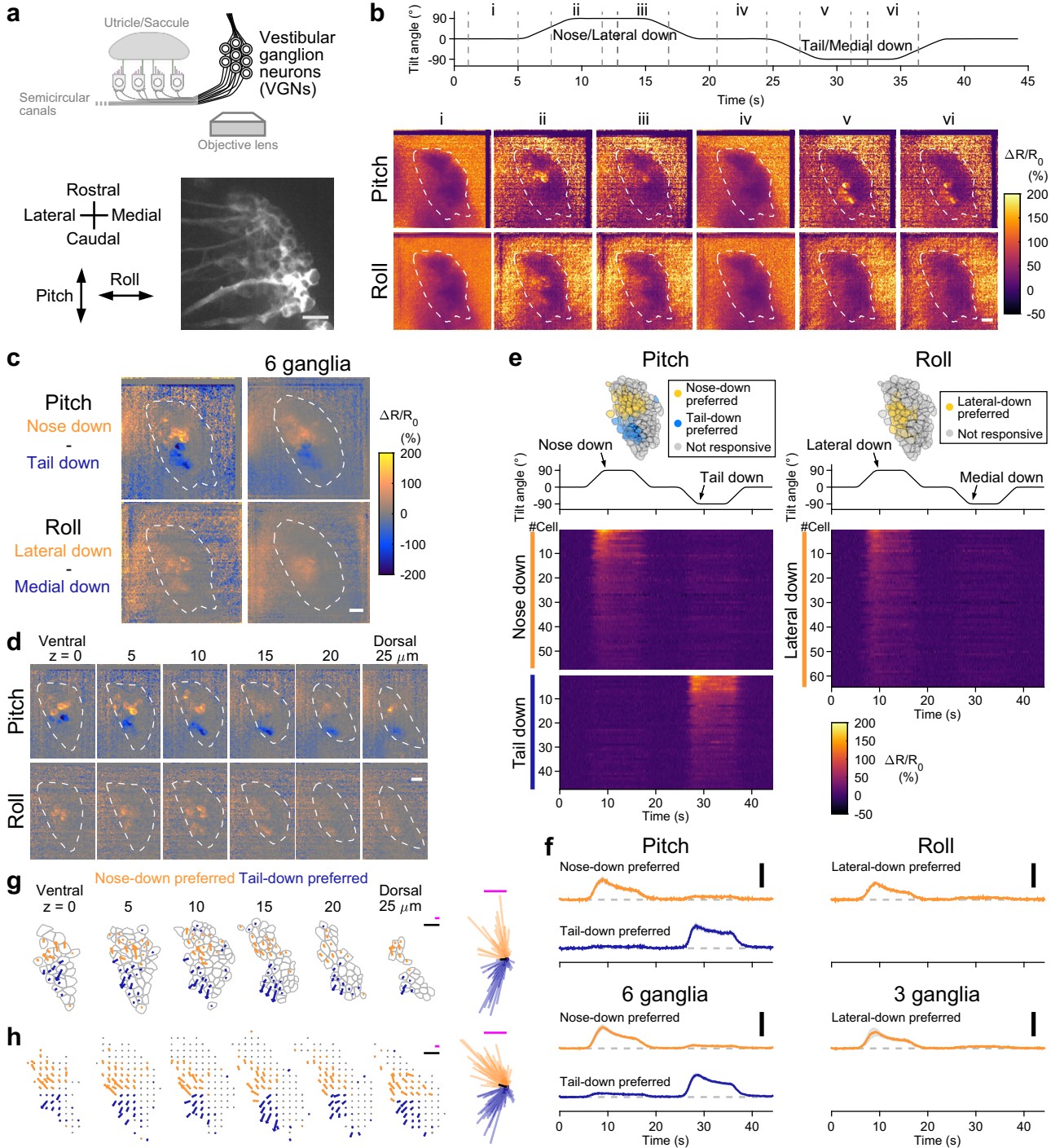

**Fig. 5 | Topographic representation of head tilt direction in VGNs. a** Imaging of VGNs (top). VGNs in the rostral division of VG in a 5 dpf *Tg(myo6b:jGCaMP7f; myo6b:tdTomato)* larva imaged from ventral (bottom). Red channel is shown. Scale bar: 10 μm. The same orientation applies to (**b**)–(**e**), (**g**), and (**h**). **b** VGN responses to 90° static tilt (see Supplementary Movie 5). Tilt time course (top). Sequential images showing mean $\Delta R/R_0$ responses during the time period "i" to "vi" (bottom). Maximum intensity projection of six optical slices with 5 μm intervals. Single trial at each slice. Dashed lines: VG contours. Scale bar: 10 μm. The same ganglion is shown in (**c**), (**d**), (**e**), and (**g**). Response selectivity for tilt direction in images of maximum intensity projection (**c**) and different depth (**d**). Tail-down response ("v" in [**b**]) subtracted from nose-down response ("ii" in [**b**]) (top). Medial-down response ("v" in [**b**]) subtracted from lateral-down response ("ii" in [**b**]) (bottom). Dashed lines: VG contours. Scale bar: 10 μm. Average of 6 ganglia ([**c**], right). **e** Tilt responses per VGN. VGN location (top, 6 optical slices overlaid). Tilt time course (middle). $\Delta R/R_0$

responses (bottom). **f** Mean $\Delta R/R_0$ responses across VGNs in each preference group in a single VG (top). Number of cells is reported in the Supplementary Fig. 4 legend. Mean responses of each group (bottom, 6 VG for pitch; 3 left VG for roll; 3 right VG are shown in Supplementary Fig. 4e). SEM is shown in gray. Tilt time course is the same as in (**e**). Scale bar: 50% $\Delta R/R_0$. **g** VGN response vectors. Orientation and length of arrows indicate preferred direction and $\Delta R/R_0$ amplitude, respectively. Responses with amplitude larger than 10% $\Delta R/R_0$ are shown. Gray: VGN contours. Magenta scale bar: 50% $\Delta R/R_0$. Black scale bar: 10 μm. (right-end panel) All response vectors aligned together. Black bar: mean response vector. Magenta scale bar: 50% $\Delta R/R_0$. **h** Summary of VGN response vectors from 6 ganglia. Mean response vectors per grid (see "Methods"). Arrows and gray dots show mean response vectors with amplitude larger or not larger than 10% $\Delta R/R_0$, respectively. Colors and scale bars are the same as in (**g**). Source data are provided as a Source Data file.

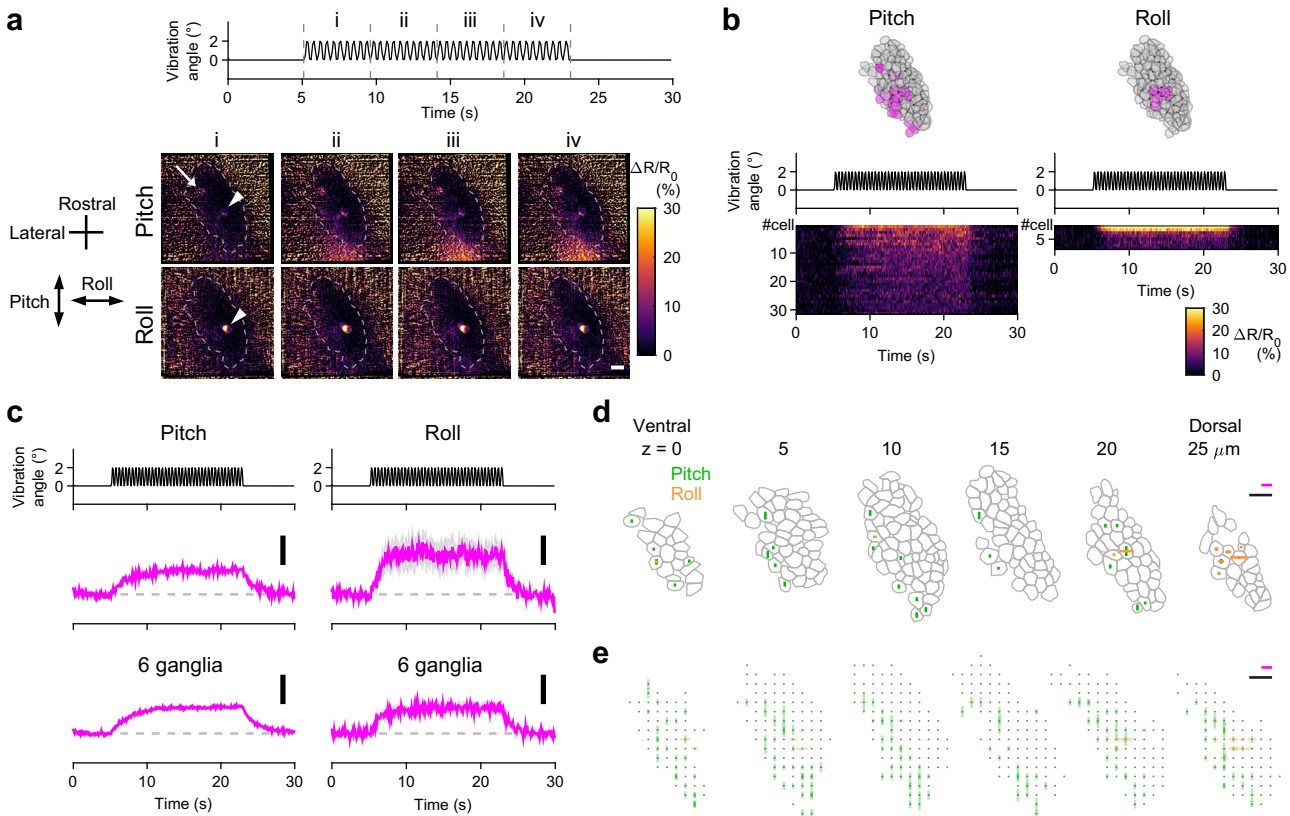

**Fig. 6 | A subpopulation of VGNs relay head vibration signals. a** Example of VGN responses to head vibration in pitch/roll axis (see Supplementary Movie 6). Vibration time course (top). Sequential images showing changes of mean $\Delta R/R_0$ during the time period "i" to "iv" from those during the pre-stimulus period (time 0 to 5 s) (bottom). Maximum intensity projection of VGNs in the rostral division of VG spanning a depth of 25 μm in a *Tg(myo6b:jGCaMP7f; myo6b:tdTomato)* larva imaged from ventral at 5 dpf. One neuron responds to both stimuli (arrowheads) whereas another responds to only pitch vibration (arrow). Dashed lines: VG contours. Scale bar: 10 μm. **b** Example of vibration responses per VGN. The same ganglion shown in (**a**). Responses to pitch (left) and roll (right) vibration. Location of activated (magenta) and not-activated (gray) VGNs (top, 6 optical slices overlaid). Vibration time course (middle). Responses ($\Delta R/R_0$) per activated VGNs (bottom). **c** Mean responses in activated VGNs. Vibration time course (top). Mean $\Delta R/R_0$ (middle). Pitch: 31 cells. Roll: 8 cells. Mean $\Delta R/R_0$ in 6 ganglia (bottom). Deviation (SEM) is shown in gray. Scale bar: 10% $\Delta R/R_0$. Example (**d**) and summary (**e**) of VGN response vectors. Magenta scale bar: 20% $\Delta R/R_0$. Black scale bar: 10 μm. **d** The same VG shown in (**a**). Bars indicate $\Delta R/R_0$ amplitude in activated VGNs. Gray: VGN contours. **e** Responses from 6 VG shown in translucent, overlaid bars (see "Methods"). Gray dots: not-activated VGNs. Source data are provided as a Source Data file.

vectors pointing to the rostral-lateral directions whereas the caudally located VGNs had vectors pointing to the caudal-lateral directions (Fig. 5g, h). Thus, in vivo Ca²⁺ imaging visualized the topographic representation of the static tilt direction in the VGNs.

## A small subpopulation of VGNs relay vibratory head motion signals

How the vibratory head motion signals are conveyed by the VGNs was examined by imaging VGN activity during vibration stimulus. In the rostral division of VG, vibration stimulus in the pitch and roll axes increased $\Delta R/R_0$ in a small number of VGNs (Fig. 6a, b and Supplementary Movie 6). The number of roll vibration–responsive VGNs was especially small. This sparse activity pattern was observed in the 6 ganglia examined. The mean $\Delta R/R_0$ of the activated neurons increased and reached a plateau during the vibration (Fig. 6c). The spatial distribution of the responses was visualized in the optical slices (Fig. 6d, e). The activated neurons spanned from the center to the lateral and to the caudal region in the VG, and the roll vibration–responsive neurons appeared to be located dorsally. Among the activated VGNs, there were neurons that responded to both pitch and roll vibration (7.9 ± 2.2%), pitch only (86.8 ± 3.4%), or roll only (5.3 ± 2.9%) (mean ± SEM, 6 VG, among the total imaged VGNs, 1.0 ± 0.3, 10.9 ± 1.1, and 0.8 ± 0.3%, respectively). Thus, the vibration signals from the HCs are sent to the brain through a small population of vibration-responsive VGNs.

Based on the responses to the static tilt and vibration, the imaged VGNs in the 6 VG were categorized into four groups: those responsive to both head tilt and vibration (tilt and vibration: 10.2 ± 1.1%); tilt only (tilt selective: 39.2 ± 4.2%); vibration only (vibration selective: 2.3 ± 0.7%); and not responsive to tilt and vibration (48.3 ± 4.4%). Taken together, the static head tilt and vibration signals are sent to the brain through tilt- and vibration-responsive, tilt-selective, and vibration-selective VGNs.

## VGNs innervate utricular HCs in a topographic manner
The spatially ordered tilt direction preference in the utricular HCs and VGNs (Figs. 2, 4, and 5) indicated that the VGNs receive inputs from the HCs in a topographic manner. To morphologically examine the pattern of HC innervation by the VGNs, we used a photoconversion method in *Tg(hspGFF53A; UAS:Kaede)* larvae that expressed Kaede in the VGNs. At 5 dpf, the green-Kaede was photoconverted to red-Kaede only in the somata of the rostral half in the rostral division of VG by a 405 nm laser (Fig. 7a). The red-Kaede diffused from the VGN somata to their neural processes. The color of the neural processes in the rostral utricle turned reddish. The spatial distribution of the red-Kaede in the neural processes in the utricle was quantified by the voxel fluorescence intensity ratio (red/green). The rostral utricle contained large red/green intensity ratio values whereas the caudal utricle contained small ratio values (Fig. 7b), indicating that the rostral VGNs projected to the

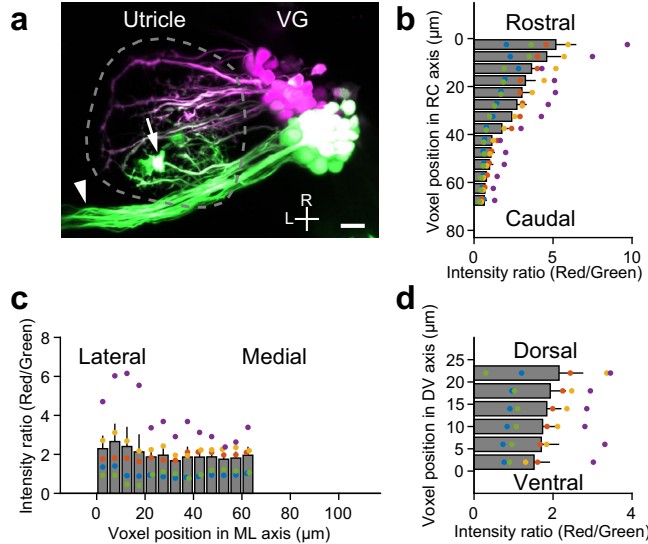

**Fig. 7 | VGNs innervate utricular HCs in a topographic manner. a** Maximum intensity projection image of VGNs expressing Kaede in a *Tg(hspGFFS3A; UAS:-Kaede)* larva at 5 dpf. Green-Kaede (green) is photoconverted to red-Kaede (magenta) only in the rostral vestibular ganglion (VG) somata. Dashed line encloses the utricle. Horizontal canal nerve (arrowhead) and ectopic Kaede-expressing cells (asterisk) overlapping the utricle are excluded from quantitative analysis. R: rostral. L: lateral. Scale bar: 10 µm. Distribution of voxel fluorescence intensity ratio (red/green) in utricles in the rostrocaudal (RC) (**b**), mediolateral (ML) (**c**), and dorso-ventral (DV) (**d**) axes. Mean and SEM from 5 utricles. Large ratio values indicate a large amount of photoconverted red-Kaede fluorescence in the voxels. Source data are provided as a Source Data file.

rostral utricle. Compared to this large gradient in the rostrocaudal axis, the red/green ratio was mostly constant in the mediolateral and dorsoventral axes (Fig. 7c, d). Thus, the rostrally located utricular HCs are innervated by the rostrally located VGNs. Similar results were obtained in another set of experiments in which green-Kaede was photoconverted in the neural processes only in the rostral half in the utricle (Supplementary Fig. 5). Together with the tilt direction–selective responses in HCs and VGNs (Figs. 2, 4, and 5), these results demonstrated that the utricular VGNs topographically represent tilt direction by receiving inputs from the utricular HCs via a spatially ordered pattern of innervation.

## Discussion

Functional imaging of a neural population at the single-cell level during natural head movement is challenging. Light-sheet $Ca^{2+}$ imaging during artificial displacement of an otolith by optical trapping[20,33] or physiological roll tilt stimulus[21] in larval zebrafish was reported. However, these methods have limitations. Optical trapping produced small otolith displacement between 15 and 140 nm. However, the displacement was an order of magnitude smaller than the HC's wide operating range (>1 µm)[14,34]. Furthermore, in order to reproduce the otolith displacement during physiological head motion, optical trapping needs to be calibrated to a measured otolith displacement during physiological head motion. However, the otolith displacement during head motion is currently unknown. In the latter study, the feasible tilt angle range was limited within ±25°, presumably because the artificial fluorescence intensity change surpasses the neural activity-related signals as the tilt angle increases. As we showed in Supplementary Figs. 2b–e and 4a–d, however, HC and VGN exhibited maximal responses to static tilt at approximately ±90°. Thus, the previous study measured responses to static tilt within a small fraction of the wide dynamic range of the vestibular sensory system. Moreover, although light-sheet microscopes enable brain-wide imaging, they do not

visualize all cells. In the ear, the otolith scatters light and prevents imaging. The location of HCs and VGNs that are ventrolateral to the brain also makes it difficult to image individual cells. Therefore, a new method—tiltable objective microscope—was required. Combined with a new in vivo preparation, the present study visualized all HC and VGN activity during head movement.

The tilt of the imaging system causes mechanical distortion in the optics due to the direction change of the gravitational force on the optics[21]. In addition, during the stage rotation, the excitation beam was static whereas the objective lens and fish rotated. This resulted in rotating bead/cell positions in the static excitation beam. Since the beam intensity profile was unlikely to be perfectly homogeneous, the excitation beam intensity at each bead/cell changed depending on the positions during the stage rotation. These factors contributed to a large artificial fluorescence intensity change during the tilt (Fig. 1e, k). By two-color ratiometric imaging, the artificial signal changes were confined within ±10% during static 360° tilt and within ±5% during vibration stimulus (Fig. 1 and Supplementary Fig. 1). These artifacts were considerably smaller than the large responses in the HCs and VGNs that exhibited ~300 and ~200% $\Delta R/R_0$ increase during the static tilt, respectively (Figs. 2d and 5e). Therefore, neural activity can be quantified by this imaging method. As a further demonstration of the usability, our further work using the tiltable objective microscopes imaged the activity of brain neurons and axial muscles during a roll tilt-induced body bend behavior and revealed the biomechanics and neural circuits of the posture control behavior in larval zebrafish[35]. Thus, the tiltable objective microscope, with static/dynamic vestibular stimulus in different axes and flexibility in the observation direction of a specimen, can be used for the functional imaging analysis of the vestibular processing.

To image HC/VGN responses to dynamic head motion, vibration stimulus with multiple frequency peaks was delivered to a specimen (Supplementary Fig. 1g). The reasons why this stimulus was used instead of widely used sinusoidal acceleration stimulus at a specific frequency were because oscillatory acceleration changes with large amplitude were required, and because the constraint of the rotation stage motion necessitated it. To produce the large acceleration, we used the angular position changes of the rotation stage in a trapezoidal waveform (Supplementary Fig. 1d). The go-and-stop motion produced a relatively larger acceleration than a sinusoidal stage movement. In exchange for this large amplitude, ringing acceleration at the higher frequencies unavoidably occurred. Also, the rotation stage does not quickly change the direction of rotation. Therefore, a series of oscillatory acceleration stimuli, each of which occurred at a specific frequency, were not produced in this study, and frequency responses of the HCs/VGNs were not examined. However, the tiltable objective microscope is modularly composed and can be combined with another stimulus device (e.g., a translational platform) that can shake the otolith organs at different frequencies. Using such a stimulus device, frequency tuning will be examined in future research.

To increase the throughput of the imaging, the imaging area can be enlarged by using another objective lens and/or confocal scanner for wide-area imaging[36]. Imaging volume can also be increased by adding an electrically driven lens positioner[37]. Since the spinning disk confocal optics can be combined with a two-photon excitation optics[38], the design of the tiltable objective microscope may also be used for in vivo imaging of less transparent tissues in other animals.

In otolith organs, hair-bundle polarity is gradually different but spatially ordered between neighboring HCs, and it reverses at the LPR. The LPR runs close to the anterior-lateral edge in the fish utricle[23,27,28]. Consistent with an inference from the polarity, $Ca^{2+}$ imaging demonstrated that the direction of head movement is indeed encoded in the spatially clustered population of HCs (Figs. 2–4). In HCs, intracellular $Ca^{2+}$ reflects both mechano-electrical transduction at the hair bundles[39,40] and vesicle release at the basal active zone[40,41]. Since the

acceleration during the vibration stimulus occurred intermittently (Supplementary Fig. 1d), the gradually increased activity during the vibration stimulus probably reflects the $Ca^{2+}$ accumulation by the repeated, intermittent gating of the $Ca^{2+}$-permeable channels including mechano-electrical transduction channels and voltage-gated channels, and it is also likely to reflect a long decay time constant of the dissociation between $Ca^{2+}$ and GCaMP[42]. Using the tiltable objective microscope, hair-bundle/otolith displacement can be imaged during the tilt and vibration stimulus. The contribution of $Ca^{2+}$ influx through L-type $Ca^{2+}$ channels[43], a prominent source of $Ca^{2+}$ entry at the basal active zone, can be examined by pharmacologically blocking the channels.

Furthermore, the present study revealed that striolar and extrastriolar HCs responded differently to the static tilt and vibration stimulus (Fig. 4). The HC responses to the static tilt reflect responses to a sustained deflection of hair bundles whereas the responses to the vibration presumably reflect oscillatory hair-bundle motion. The labeled region in a transgenic fish (Fig. 4a, b) was slightly broader than the striola characterized in electron microscopy (EM) data[23]. Therefore, the striolar HCs analyzed in the present study may also contain peristriolar HCs. Nevertheless, a clear difference was found: static tilt was preferentially, though not exclusively, received by the extrastriolar HCs whereas vibratory head motion was preferentially transduced by the striolar HCs. The EM data also showed that VGNs with myelinated axons tend to innervate striolar HCs whereas those with unmyelinated axons tend to innervate extrastriolar HCs. Thus, the decomposition of static/dynamic linear acceleration in the spatially segregated HC populations leads to signal transmission to the brain through the two channels of the VGNs.

The two channels are also morphologically characterized by distinct afferent terminals in amniotes; the striola contains more type I HCs innervated by calyx-shape afferent terminals whereas the extrastriola contains more type II HCs innervated by bouton-shape terminals. In fish, in contrast, all HCs are classified as type II[23]. This suggests that fish have a basal, relatively simple form of vestibular periphery that originated in the common ancestor of vertebrates for decomposition of the different temporal dynamics of head motions.

The different responses between the striolar and extrastriolar HCs may be attributed to either or combination of the following mechanisms; fish data is relatively sparse and therefore possibilities are discussed largely based on studies in frogs, turtles, and rodents. First, stiffness of the hair bundles, which provides a large fraction of reactance to shearing motions of otolith/otolithic membrane[44], systematically differs across the macular regions[45]. The higher stiffness in the striolar HCs likely underlies the larger responses to the dynamic, vibratory stimulus, which is consistent with the fact that sound-recipient cochlear HCs have a high stiffness. Second, hair-bundle shape and mechanical coupling between otolith and hair bundles differ across striolar and extrastriolar regions, which is thought to tune HCs to different stimulus frequency ranges[9,23,46–52]. Third, the mechano-electrical transduction currents per unit hair-bundle deflection differ between HCs[53,54]. Fourth, in striolar HCs, mechano-electrical transduction adaptation and voltage-dependent electrical tuning contribute to the high-pass filtering[55,56].

Since the intracellular $Ca^{2+}$ level affects membrane potential via $Ca^{2+}$-dependent $K^+$ channels, $Ca^{2+}$-binding proteins, which are abundant in the striolar region (Fig. 4), may regulate the electrical tuning by buffering intracellular free $Ca^{2+}$. Consistent with this, $Ca^{2+}$-binding proteins are richly expressed in most regions in the saccule, which receives vibration and sound[30]. With the tiltable objective microscope, saccular responses to vibration and static tilt can be imaged in future study. Whether the binding between $Ca^{2+}$ and GCaMP affects the HC's functional properties is currently unknown. However, mechano-electrical transduction and vesicle release were observed by $Ca^{2+}$ imaging[39–41], and GCaMP-expressing fish were indistinguishable from non-expressing siblings, which suggests that GCaMP had minor effects, if any.

Otolith VGNs relay a wide range of direction signals. The present study demonstrated that the different tilt direction signals are represented in the spatially ordered afferent pathway from the HCs to the VGN somata (Figs. 5 and 7). This functional direction selectivity map is consistent with an inference from the EM data in which the direction preference of VGNs was estimated from synaptic connections with utricular HCs[23]. The location of vibration-responsive VGNs (Fig. 6) is also largely consistent with the position of striola-innervating VGNs in the EM data. Since the basic structure of the vestibular system is thought to be largely conserved in vertebrates, the other vertebrate species also probably have a similar, spatially ordered circuit organization. The number of VGNs that did not exhibit responses to tilt/vibration stimulus was not small. The rostral division of the VG receives inputs not only from the utricle but also from the saccule and the anterior and horizontal semicircular canals. Although reading out the neural activity by $Ca^{2+}$ imaging might have overestimated the number of non-responsive VGNs, these VGNs are likely to respond to linear acceleration in the dorsal-ventral axis (which the saccule receives) or, after the formation of the semicircular canals in developed fish, angular acceleration (which the semicircular canals receive).

During static roll tilt, many lateral-down preferred VGNs were observed whereas medial-down preferred VGNs were rarely found (Fig. 5). This is in line with previous studies; most inferred afferents and vestibulospinal neurons are tuned to the lateral-down direction[22,57]. Thus, the static roll tilt inputs to the brain are mostly derived from the downside ear through the lateral-down preferred VGNs in larval zebrafish. The EM data, however, indicated that a small number of medial-down preferred VGNs exist[23]. These VGNs form synapses with striolar, extrastriolar, and developing HCs located laterally to the LPR. One possible reason for this apparent discrepancy is that, in this larval period, these VGNs may be functionally immature. Artificial fluorescence intensity change during tilt might have masked the nascent responses. Another possibility is that the medial-down preferred VGNs respond to phasic inputs rather than static tilts. Consistent with this, a few VGNs were tuned to the medial-down phasic acceleration[22]. If the latter is the case, roll-vibration-responsive VGNs (Fig. 6) may include the medial-down preferred, phasic VGNs.

VGNs encode the intensity and direction of acceleration by increasing/decreasing the firings from the resting discharge rate; in mammals, up to ~100 Hz in both regular afferents, which spontaneously fire with regular inter-spike intervals, and irregular afferents, which discharge with varying intervals[58]. $Ca^{2+}$ imaging visualized that a VGN subset increased $\Delta R/R_0$ during the static tilt to the preferred direction whereas the tilt to any direction did not evoke negative $\Delta R/R_0$ changes (Fig. 5). In larval zebrafish, the spontaneous firing rate does not appear markedly low[22,57], therefore decline in the VGN firing rate during tilt was supposed to decrease the fluorescence ratio. The cause of the absence of negative responses in the imaging is unclear. One possibility is that, compared to the HCs where $Ca^{2+}$ continuously enters through large conductance channels that are open in the resting state, the intracellular $Ca^{2+}$ level in the VGNs in the resting state may not be high. Even when VGNs decrease the firing rate during tilt, decrease in the fluorescence ratio might have been masked by the artificial fluorescent intensity changes.

Otolith VGNs also convey vibration signals. Previous studies showed that striola-innervating irregular afferents, the number of which is smaller than extrastriola-innervating afferents, relay the dynamic head motion signals in mammals[6,7,11]. Similarly, anatomical data in larval zebrafish showed that a relatively small number of VGNs innervate the striolar HCs[23]. Consistent with these data, the number of vibration-responsive VGNs, especially those responsive to the roll axis, was small (Fig. 6b, d and e). The difference in the number of roll and pitch vibration–responsive VGNs presumably reflects the

contribution of the saccular inputs to the pitch vibration–responsive VGNs; the saccule is supposed to respond to head motions in the pitch but not the roll axis[32]. In guinea pigs, striola-innervating utricular and saccular irregular afferents with low spontaneous discharges respond to high-frequency, bone- and air-conducted vibrations[59]. The vibration cues to the brain are thought to be used to detect one's motion in humans[60].

As for the spatial signal processing, in the visual and somatosensory systems and auditory brainstem, the spatial identity of a stimulus is maintained in the spatially ordered neural circuits[61]. In the vestibular system in larval zebrafish, spatial segregation of direction-preference pathways has been anatomically reported in vestibulo-ocular neurons in the vestibular nuclei[23]. The directional head movement signals visualized in the present study are probably sent to these vestibular nuclei in a spatially ordered manner.

In vertebrates, tonic and phasic head movement signals converge on overlapping regions in the vestibular nuclei, where most second-order neurons receive a mixture of these signals[22,62–64]. Subgroups of the second-order vestibular neurons with different membrane properties and discharge patterns are suitable for processing the phasic or tonic signals[65]. Although substantial progress has been made in understanding the vestibular system, including the vestibulo-ocular circuits, how the temporally different signals are processed in the brain to produce context-appropriate behaviors remains ambiguous. The tiltable objective microscope enables functional imaging in the brain at the single-cell level[35]. Thus, imaging analysis during physiological head movement with the new microscope will be a powerful approach to obtain a deeper understanding of the circuit mechanisms of vestibular processing.

## Methods

### Animals

Transgenic zebrafish lines, *Tg(myo6b:jGCaMP7f; myo6b:tdTomato)*, *Tg(myo6b:jGCaMP7f; s100s-hs:tdTomato)*, and *Tg(hspGFF53A; UAS:Kaede)*, were used. The transgenic fish were indistinguishable from wild-type siblings, matured as adults, and produced healthy offspring. All zebrafish larvae used for experiments were at 5 dpf. Sex is not yet determined at this developmental stage. Under visual inspection, all transgenic larvae were indistinguishable from the wild-type siblings. Therefore, the transgenic larvae were screened based on the fluorescence expression. All experiments and procedures were approved by the animal care and use committees of the National Institutes of Natural Sciences and comply with ARRIVE guidelines. Fish were raised and maintained in the Center for Animal Resources and Collaborative Study facility at 28.5 °C with a 14:10 light and dark cycle. Larval zebrafish were housed in Petri dishes with fish-rearing system water. *Tg(myo6b:jGCaMP7f)* was generated in this study by Tol2 transposon-mediated transgenesis[66]. *myo6b* promoter[5] and jGCaMP7f[42] sequences were inserted in this order into a pT2KXIGΔin vector. *Tg(s100s-hs:tdTomato)* was generated in this study using a CRISPR/Cas9-mediated knock-in technique[31]. Donor plasmid was generated by inserting Mbait, hsp70 promoter, and tdTomato sequences into a plasmid DNA. Genome DNA sequence for targeting *s100s* by short guide RNA was 5′-GGCCATTTC ACACTGCTCCA-3′. *Tg(myo6b:tdTomato)*[29], *Tg(hspGFF53A)*[67] and *Tg (UAS:Kaede)*[68] were generated in previous studies. Transgenic larvae with a transparent *mitfa*[−/−][69] pigment mutant background were used for experiments.

### Design and setting of the tiltable objective microscope

The tiltable objective microscope consisted of an objective lens unit, rotation stage unit, tube lens unit, spinning-disk confocal scanner, image splitting optics unit, camera, and laser. Key optomechanical components, software, and algorithms are listed in Supplementary Tables 2 and 3. In the rotation stage unit, a stage adaptor plate was fixed on a motorized rotation stage. The objective lens unit, rotation stage unit, and tube lens unit were connected via metal rods of a ThorLabs' 30/60 mm cage system with cage adaptor plates. A specimen chamber was fixed on a rotation mount with dental utility wax (100 g red, GC Corporation), and the specimen position was set by an XY translation mount. For fish imaging, fish yawing orientation was set by the rotation mount. The fish orientation specified the axis (pitch or roll) of the stimulation when the motorized stage rotated. Imaging plane was set by a lens positioner. After the lens position was set, the lens was fixed by screws on a cage mount. In order to excite green or red fluorescent proteins or beads, a 488 nm excitation laser beam was delivered to the specimen through the confocal scanner, tube lens unit, rotation stage unit, and an objective lens (20×, NA 0.8 for beads or 40×, NA 0.8 for fish). In the objective lens unit, a 45° mirror (elliptical mirror on a kinematic mirror mount) deflected the excitation and emission light paths 90°. The laser power was 0.5 mW/mm² with 20× and 1.7 mW/mm² with 40× objectives on the specimen. The excitation and emission light were separated by a dichroic mirror in the confocal scanner. The emission light was split to green and red spectra by another dichroic mirror and independently filtered by the emission filters in the image splitting optics unit. The green and red emission light formed images on an image sensor in the camera. The microscope model was drawn with FreeCAD 0.18.

The fluorescence images were recorded with HCImage Live software using the W-view mode. Image binning size was 1 for imaging with the 20× objective lens and 2 for imaging with the 40× lens. Image resolution was 3.4 pixels per micrometer for both conditions. The green and red channels were vertically combined in the acquired images. The image size per channel was 448 × 456 pixels for beads and hair cell imaging and 320 × 316 pixels for SAG imaging. The images were acquired at 10 frames/s with 16-bit depth. To synchronize the timing of the stage rotation and the image acquisition, a voltage pulse was generated in the stage controller at the beginning of a stimulus session and sent to the camera.

### Static tilt and vibration stimulus

The motorized rotation stage was driven by a stage controller and controlled by Kinesis software. For the 90° tilts, the stage was still for 5 s and rotated 90°, then the stage position was kept for 5 s and moved back to the original position. After keeping the still position for 5 s, the stage was rotated −90°, then the stage position was kept for 5 s and moved back. The maximum rotation velocity was 25°/s with angular acceleration of ±24.9°/s². The stage rotation angle was calculated from the fluorescent beads images. For the 360° tilts, the stage was still for 3 s and rotated 360°, then the stage position was kept for 6 s and rotated −360°. The rotation was driven with the maximum rotation velocity of 15°/s with angular acceleration of ±24.9°/s². The angular acceleration was selected to minimize the oscillation of the stage at the onset and end of the stage movement. For the vibration stimulus, the stage was bidirectionally rotated at 2° angle changes with a trapezoid waveform in position at 2.2 Hz (Supplementary Fig. 1g) with the maximum rotation velocity of 1000°/s and angular acceleration of ±1000°/s² for 18 s. The stage movement during the vibration stimulus was imaged by a digital camera (acA640-750um, Basler) at 500 frames/s to calculate the stage rotation angle. Inertial acceleration during the stage movement was measured by a 3-axis accelerometer (MA3-04AD, MicroStone) and sampled at 1 kHz with a digitizer (Digidata 1440A, Molecular Devices) and software (pClamp10, Molecular Devices).

### Fluorescent beads imaging

Fluorescent beads with a diameter of 1 μm were placed on a glass sheet (D11130H, Matsunami). The glass was fixed on the objective lens unit with dental utility wax. The unit was set on the tiltable objective microscope such that the beads on the glass were imaged through the glass; the objective lens faced upward like an inverted microscope. The

beads were imaged during the stage rotation at three different fields-of-view (FOVs).

## Fish specimen preparation

A larva at 5 dpf was immobilized with 1 mg/mL alpha-Bungarotoxin (Biotium, #00010-1). In Kaede-expressing fish imaging, the larva was transferred to an acrylic specimen chamber (inner dimension: 14 mm × 14 mm × 1 mm) and embedded dorsal up in 2.0% low melting point agarose (ThermoFisher Scientific, 16520050) dissolved in fish-rearing system water. After the agarose solidified, the larva was covered with fish-rearing system water and partially photoconverted by UV light. In the other experiments, the immobilized larva was transferred to a 0.02% tricaine methanesulfonate (Sigma-Aldrich, A5040) anesthetic solution dissolved in an extracellular solution containing (in mM) 134 NaCl, 2.9 KCl, 1.2 MgCl$_2$, 2.1 CaCl$_2$, 10 HEPES, and 10 glucose, adjusted to pH 7.8 with NaOH, and the ventral portions of the head, including the jaw, operculum, and heart, were removed using a pair of forceps. After the dissection, the larva was transferred to the specimen chamber and embedded ventral up in 2.0% low melting point agarose dissolved in the extracellular solution. After the agarose solidified, the larva was covered with the extracellular solution.

The chamber containing the fish was tightly sealed by an acrylic lid (20 mm × 20 mm × 2 mm with a 12 mm-diameter center window) with silicon pads (Pillow Soft #7, Mack's). The center window of the lid was a sheet of transparent fluorinated ethylene propylene (FEP), through which the fish was imaged. The chamber was fixed on the objective lens unit with dental utility wax. The space between the water-dipping objective lens (LUMPLFLN40XW, Olympus) and the FEP sheet was filled with distilled water. An O-ring (inner diameter: 10.8 mm; width: 2.4 mm) attached to the chamber lid by silicon pads prevented the water from dripping off while the objective lens unit tilted. The objective lens unit with the chamber was set on the tiltable objective microscope such that the fish was oriented dorsal up in the resting position. The fish was imaged from the dorsal for the Kaede-expressing fish; the objective lens faced downward like an upright microscope. In the other experiments, the fish was imaged from the ventral; the objective lens faced upward like an inverted microscope. The same rotation sequences that were used for the beads imaging were reused for the offline registration of the rotated images.

## Imaging Kaede-expressing fish during static tilt and vibration stimulus

Transgenic larvae *Tg(nefma-hs:Gal4; UAS:Kaede)* were used. The green- and red-Kaede-expressing vestibulospinal and reticulospinal neurons, neurons in the oculomotor and trochlear nuclei, and neurons in the nucleus of the medial longitudinal fascicles in a partially photo-converted larva was imaged during the stage rotation.

## Imaging HCs during static tilt and vibration stimulus

Transgenic larvae *Tg(myo6b:jGCaMP7f; myo6b:tdTomato)* and *Tg(myo6b:jGCaMP7f; s100s-hs:tdTomato)* were used. An optical slice that contained as many utricular HCs as possible was imaged during the stage rotation in the pitch or roll axis. After the recording, the same HCs were imaged in the same optical slice during the stimulus in the orthogonal axis.

## Imaging VGNs during static tilt and vibration stimulus

Transgenic larvae *Tg(myo6b:jGCaMP7f; myo6b:tdTomato)* were used. Neurons in the rostral division of the VG were imaged at 6 optical slices with 5 μm intervals. The neural activity was imaged per slice during the stage rotation in either the pitch or roll axis; images were recorded by a series of single-optical-slice imaging, not by a volumetric scan, during a series of 90° tilt, 360° tilt, and vibration stimulus. After the recording, the same VGNs were imaged in the corresponding optical slices during the stimulus in the orthogonal axis.

## Morphology of a utricle in transgenic fish

Transgenic larvae *Tg(myo6b:jGCaMP7f; s100s-hs:tdTomato)* were used. A larva was dissected as described above. The fish was embedded dorsal up in 2.0% low melting point agar dissolved in the extracellular solution on a glass-bottom dish (D11130H, Matsunami) and covered with the extracellular solution. An optical slice of the utricle was imaged from the ventral by an inverted confocal microscope (TCS SP8 MP, Leica) using an objective lens (HC PLAPO 40×/1.10 W CORR CS2, Leica) and LAS X software (Leica) with *xy* resolution 0.190 μm/pixel. Laser lines of 488 and 552 nm were simultaneously illuminated. Fluorescent signals were simultaneously acquired by two detectors through wavelength selectors (emission spectra: 495–545 nm for jGCaMP7f; 562–700 nm for tdTomato).

## Photoconversion and imaging of the VGNs

Transgenic larvae *Tg(hspGFF53A; UAS:Kaede)* were used. A Kaede-expressing larva was dissected as described above. The procedure for the fish embedding in agarose and microscopy was the same as in 'Morphology of a utricle in transgenic fish' section. Kaede-expressing neuron somata in the rostral half of the rostral division of the VG or peripheral neuronal processes in the rostral utricle were selectively photoconverted by a 405 nm laser with a zoom factor between 10× and 20×. After the photoconverted red-Kaede diffused from the somata to the peripheral neural processes or from the processes to the somata, a Z-series image stack containing the utricle and the ganglion was acquired with 0.5 μm depth steps with *xy* resolution of at least 0.142 μm/pixel. Laser lines of 488 and 552 nm were simultaneously illuminated to excite green- and red-Kaede. Fluorescent signals were simultaneously acquired by two detectors through wavelength selectors (emission spectra: 495–545 nm for green-Kaede; 562–700 nm for red-Kaede).

## Acceleration during static tilt and vibration stimulus

Two orthogonal vector components of the gravitational acceleration (**g**, 1.0 **g** = 9.806 m/s$^2$), one in the centripetal direction and the other in the tangential direction, were simulated from the rotation angle (θ): **X**(centripetal) = **g**· cosθ; **Y**(tangential) = **g**· sinθ. Since the inertial acceleration measured by the accelerometer was prominent only in the tangential direction during the stimulus, the signal in this axis was processed by the fft function in MATLAB to visualize frequency components.

## Image registration

Data was analyzed in MATLAB (R2019b, MathWorks) with the Image Processing Toolbox. Three-dimensional (*x*, *y*, *t*), raw time-series images contained green and red fluorescence signals in the top and bottom half, respectively. The two-color channels were manually aligned and processed as a four-dimensional data matrix (*x*, *y*, *t*, channels). The image sensor in the camera contained a small number of 'hot pixels' where pixel intensity was a constant high value (3 pixels in 448 × 456 pixels FOV, 0.0015%). The hot pixels were prominent as bright arcs when image rotation was corrected. To avoid using the hot pixels, the intensity in the hot pixels was replaced by a median of surrounding pixels in a 3 × 3 pixel window. The recorded images rotated while the objective lens unit tilted. The image rotation angle during the tilt, which was equal to the tilt angle of the specimen, was calculated by the beads position for every image frame in the beads imaging data. Using the image rotation angle data, the images were registered by counter-rotating images. To keep the image rotation angle consistent across experiments, the same tilt and vibration stimulus was used in the beads and fish imaging. Residual minor image drift was registered by calculating and correcting image transformation with the imregtform and imwarp functions in the Image Processing Toolbox followed by the dftregistration function[70] at the level of 1/10 pixel. Image background intensity value, which corresponded to the first percentile in the non-

fluorescent region in the recorded images, was subtracted from all image frames. In order to align the images recorded from different organs/fish, images recorded from the right ear were flipped horizontally. The images were manually translated such that the lateral edge of the macula or VGN aligned to each other.

## Pixel-wise image analysis

Rotation-corrected, registered, background-subtracted time-series images were processed by a median filter with a window size of 3 × 3 pixels every frame. Green/red fluorescence intensity ratio images were produced pixel-wise. The mean of the ratio image frames before the stage rotation started was defined as the baseline ratio image except where otherwise noted. The relative change of the ratio images to the baseline ratio image, $\Delta R/R_0$, was calculated.

## Image analysis per bead/cell

In the fluorescent beads image analysis, mean pixel intensity within a 1.5 μm radius from the center of solitary beads was measured every image frame in the green and red channels. The green-to-red ratio of the mean intensity was calculated. Beads that aggregated with each other were manually excluded from the analysis. In the analysis of individual cells, regions-of-interest (ROIs) were drawn on each cell in ImageJ. The mean fluorescence intensity within each ROI in green and red channels was measured every image frame in MATLAB. The green-to-red ratio of the mean intensity was calculated. In the analysis of beads and cells, the mean ratio before the stage rotation started was defined as the baseline ratio. The relative change of the ratio to the baseline ratio, $\Delta R/R_0$, was calculated and smoothened by a box-car moving average of 3 consecutive time frames. Intensity of green and red signals in Kaede-expressing cells slightly changed by bleaching/photoconversion during imaging, In the analysis, the bleaching/photoconversion-induced intensity changes, which occurred largely linearly over time and independently from the stage rotation, were corrected before calculating the green-to-red ratio. The correction was carried out by subtracting the linearly changing "trend lines" from the green and red signals. To create the trend lines, signals during periods before the initial stage rotation started and after the last stage rotation ended were separately fitted by straight lines. The trend lines between these periods were estimated by a linear interpolation using the data points before the initial stage rotation and after the last stage rotation.

The amplitude of the responses to ±90° tilt was quantified as follows. Since $\Delta R/R_0$ in HCs remained largely constant while the tilt angle was kept at 90° or −90°, the mean $\Delta R/R_0$ during this period was used to determine the direction selectivity and response vector of the HCs. Since $\Delta R/R_0$ in VGNs decayed while the tilt angle was kept at 90° and −90°, the mean $\Delta R/R_0$ during the time frames from 2 s before to 2 s after the tilt angle reached 90° and −90° was used to determine the direction selectivity and response vector of the VGNs. In the VGNs, responses from the off-focus depth slightly bled through the imaging slice. Therefore, direction selectivity of VGN neurons was determined based on whether the response amplitude to one tilt direction was more than twofold larger than that to the opposite tilt direction. In both HCs and VGNs, the amplitude of the responses to vibration stimulus was quantified by mean $\Delta R/R_0$ during the last 4.5 s in the vibration stimulus. Based on the amplitude of the artificial fluorescent intensity ratio changes (Fig. 1 and Supplementary Fig. 1), cells were regarded as 'responded' to a stimulus if the $\Delta R/R_0$ was larger than 10% for tilt stimulus or 5% for vibration stimulus.

For tilt responses, a response vector of each HC/VGN was produced by the vector sum of the pitch and roll tilt responses (Figs. 2f, g and 4c). The response vector orientation represents the preferred/anti-preferred direction. The response vector length represents the square root of the sum of the squares of the $\Delta R/R_0$ amplitude during pitch and roll tilt. For vibration responses, response vectors for pitch

and roll vibration responses were individually shown as bars (Figs. 3e, f and 4d). The length of the vector sum of the bars was calculated by the square root of the sum of the squares of the $\Delta R/R_0$ amplitude in the pitch and roll vibration responses (Fig. 4h). In order to calculate mean response vectors from different organs/fish, an imaginary grid was prepared on the utricle and VG with an inter-grid interval equal to the average minimum distance between center positions of HCs (5.2 ± 0.036 μm, mean ± SEM, 510 cells from 6 utricles) and VGNs (4.1 ± 0.024 μm, mean ± SEM, 1538 cells from 6 ganglia). Each response vector was assigned to one of the crossing points in the grid that was closest to the cell position. Mean response vectors were calculated per crossing point in the grid. Since the vibration-responsive VGNs were sparsely distributed (Fig. 6), all response vectors were overlaid with 30% transparency instead of being represented by mean response vectors.

## Morphological analysis of photoconverted VGNs

Images were processed by a three-dimensional Gaussian filter with a 5 × 5 × 5 window using the imgaussfilt3 function in the Image Processing Toolbox in MATLAB. ROIs were drawn in ImageJ. In the VGN somata photoconversion data, an ROI enclosing the utricle was drawn. Voxels within the utricular ROI were thresholded to determine the voxels containing the VGN peripheral neuronal processes. In each voxel, the red and green fluorescent intensity was measured and the red-to-green ratio was calculated. A few Kaede-expressing cells, which were not VGNs, were found beneath the utricle. These cells were excluded from the analysis. In the peripheral fiber photoconversion data, ROIs were drawn in the individual VGN soma on the brightest optical section. Mean intensity within each ROI was measured for green and red channels, and the red/green ratio was calculated.

## Statistics

All statistical analysis was performed in Python 3.7 with the SciPy 1.5.2 tool[71]. Statistical methods, number of samples, and $p$ values are described in the figure legends.

## Reporting summary

Further information on research design is available in the Nature Portfolio Reporting Summary linked to this article.

## Data availability

The datasets that support the findings of this study are available in Zenodo with the identifier https://doi.org/10.5281/zenodo.7147882, and. Source data are provided with this paper.

## Code availability

Code used for registering rotated images are available in Zenodo with the identifier https://doi.org/10.5281/zenodo.7147882.

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

## Acknowledgements

The authors are grateful to Dr. Takeo Katsuki for helping the prototype design of the tiltable objective microscope. The authors are grateful to Dr. Teresa Nicolson for sharing *Tg(myo6b:tdTomato)*, to Dr. Koichi Kawakami for sharing *Tg(hspGFF53A)* transgenic zebrafish, and to Dr. Katie Kindt for sharing m6b:GCaMP6s-CAAX plasmid. The authors would like to thank Dr. Yoichi Oda for critical reading of the manuscript. Images in Figs. 4, 7, and Supplementary Fig. 5 were acquired by microscopes in the Spectrography and Bioimaging Facility, National Institute for Basic Biology Core Research Facilities. The authors thank Higashijima lab members for fish care and discussion. M.T. is grateful to the Biology of the Inner Ear course in the Marine Biological Laboratory for background knowledge on the inner ear. This work was supported by the Japan Society for the Promotion of Science, KAKENHI Grant Numbers JP18KK0215 and JP19H03333 to S.H., and JP20K06866 to M.T.

## Author contributions

M.T. and S.H. conceived the project. M.T. built the microscope, performed the experiments, and analyzed the data. I.W. and S.H. generated transgenic lines. M.T. wrote the manuscript with feedback from S.H.

## Competing interests

The authors declare no competing interests.
