## [Peer Review File · Nature Communications]

Tilttable objective microscope visualizes selectivity for head motion direction and dynamics in zebrafish vestibular systemREVIEWER COMMENTS

Reviewer #1 (Remarks to the Author):

The manuscript by Tanimoto is an exemplary piece of work using a novel optical apparatus and clever stimulus paradigms to address a long-standing question in the field: how do hair cells (and sensory afferents) respond during head motion. Admittedly, given that the mechanisms of hair cell force transduction are well established, I think that there is a strong prediction for how these cells should respond. However, while the manuscript provides the expected answer, its real value is in establishing the technology to holistically and comprehensively monitor the peripheral vestibular system. The authors report intriguing topography and differential responses to stimulation at different frequencies. I have a number of suggestions for the authors that I detail below that I think would make the manuscript more accessible and/or would reach a broader audience. I also detail a number of experimental suggestions but I stress: these are entirely optional and likely beyond the scope of the paper. I trust that the editor can review the authors' responses to my suggestions and do not need to see the manuscript again — it's nearly ready for publication. I congratulate the authors on an excellent bit of work and sincerely hope that they continue to use their apparatus and acumen to address open questions in the vestibular field. I would also encourage the authors to consider posting this to bioRxiv; we have multiple manuscripts upcoming that would benefit from being able to cite this lovely work.

David Schoppik, Ph.D.

Major suggestions:

1. Forces on the hair cells are a function of the motion of the utricular otolith. This new apparatus offers a unique opportunity to directly measure the motion of the otolith in response to linear acceleration. Similarly, it might be possible to measure the hair cells themselves. Such data would be of considerable interest to groups that model the utricle, such as Jong-Hoon Nam at the University of Rochester. It might help to shed light on later questions (see below) to know if the utricle moves with the stage, and the hair cells move with the stage, or if there are non-linearities that make this challenging. If I understand the setup correctly this should be achievable by slight changes to the geometry to allow visualization of an otic capsule mounted lateral to the objective. Again, this is just a suggestion; the results of the paper are unlikely to change with these measurements.
2. I found the differentiation of the two stimuli a bit confusing. If I understood the stimulus paradigm correctly, the biggest differentiator wasn't the temporal properties (i.e. low-pass vs. high-pass) but instead whether cells responded to large DC offsets and/or whether they responded to ~2Hz oscillations. One experimental suggestion would be to examine the responses (of hair cells, but perhaps more interestingly of VGNs) to the 2.2Hz stimulation *when the fish was at an eccentric angle*. We have observed [unpublished] that zebrafish can perform a vestibulo-ocular reflex in response to stimulation comparable to the 2Hz trapezoid at baseline and at more eccentric angles so presumably the nervous system can differentiate high-frequency oscillations even if they are superimposed on eccentric body postures.
3. The vestibular literature is quite vast and so the authors may have chosen not to address the following areas for brevity's sake. Nonetheless, I think it would be of value to discuss the following: Ian Curthoys' work examining the responses of the utricle to high-frequency vibrations in guinea pigs, Dan Merfeld's work looking at vibration discrimination in humans, and the Beraneck/Straka work examining different frequency channels in the frog central vestibular projection neurons. I strongly encourage the editor to allow the authors sufficient space to address this recent vestibular work as I feel it would expand the impact of this manuscript.

Minor comments, line-by-line

The title is a bit challenging as it isn't clear how the paper addresses "discrimination," which usually has a specific meaning to sensory physiologists.

12: "on" should be "about"

16: "visualizes" should be "reveals"

31: "received" should be "sensed"

34: the manuscript only uses the acronym MET sparsely, consider spelling it out each time.

42: Have you considered that the macula itself may not be uniform with respect to its stiffness, imparting frequency sensitivity? That may have been treated in the literature, but I am not aware of any papers in zebrafish. also 496-501

53: The word "modality" isn't quite correct — they are the same sensory modality (linear acceleration). This comment applies elsewhere e.g. 391

61: "enable" should be "enables"

78: it would be worth noting that because the fish is off the axis of rotation, any stimulus in this apparatus will have both a linear and angular acceleration.

92: reference 24 would have only treated yaw head movements, but these are pitch and roll, correct?

139: why is there a bigger change at 360 relative to 0? does that speak to the image processing?

157: it's up to the authors, but I find "nose-up" and "nose-down" to be clearer than nose/tail.

183: the arrow of the vector only points in the preferred direction, why refer to the "anti-preferred?"

213: I'm not entirely sure that figure e adds. This is the collective response of the end organ. If we are meant to compare response magnitudes across directions, why not quantify these data and show it?

213: similarly, why are the average responses comparable given the distribution of hair cells preferred directions is not. Or are these just the average responses *of cells tuned for that direction?*

248: I was astounded that there's really no response to the first 5 seconds of this stimulus, particularly since there is a response in the VGNs. It would be really nice to know what's going on. I appreciate the Discussion in 478-479 but it would be really helpful to propose a way to test your hypothesis.

263: it would be helpful to have the anatomical axes labelled in this graph (R/C and M/L)

438: do you ever see striolar/extrastriolar differentiation of axons? I ask because in 488/489 you are effectively proposing that striolar/extrastriolar are carried by different VGNs but the data don't really speak to that, do they?

515: I'd add "inferred" before afferents. You could look at Hamling et. al., still on bioRxiv for work here from our lab in support of the qualifier.

531: It's true that the vestibulospinal basal rate is low, but the rate of the inferred afferents is quite high, again, see our work on bioRxiv and Liu et. al.. So I do not think that the rate of the afferents is what is contributing to the absence of negative deltaR/R0

540: Currently, vestibular implants target the semicircular canals because they are straightforward to map; I do not believe anyone is proposing to use topography because they aren't targeting the utricle.

549: this would be a perfect place to look at the Beraneck & Straka work.

612: is the trapezoid in position? in velocity? in acceleration?

654: I sincerely hope you will do it with GCaMP next time!

791: Do consider just uploading the data to Zenodo. It's free and easy to do.

Reviewer #2 (Remarks to the Author):

General thoughts:

The authors design and build a spinning disc confocal microscope in which the stage can be rotated or shaken to produce vestibular stimuli during calcium imaging. Using this microscope they describe responses (to static and vibrational pitch and roll stimuli) in utricular hair cells that are expected based on previous descriptions of the cells' anatomical orientations. They also show that extrastriolar HCs preferentially report on static rotations while striolar HCs respond more strongly to vibrational stimuli.

In the vestibular ganglion (VG), a relatively small number of neurons show selective tilt responsiveness, and a smaller number show weak roll stimuli for lateral down specifically. These

are distributed in characteristic positions within the VG, suggesting spatial organization. Similar observations hold true for vibrational stimuli, although with lower numbers of responsive cells, especially for roll stimuli. Finally, they use targeted photoconversion to describe the anatomical innervation patterns that provide the spatial response patterns described for different stimuli in the VG.

The work is clear, the data (once ratiometric calculations are included) are convincing, and the interpretations are sound. Figures are very clear and generally convincing. There are some very minor grammatical issues, but the writing is generally very good.

Aside from a few minor comments included below, I believe that this work is already at publication quality, but I have concerns about the impact of the work on the field. There is technical impact in the development of a new type of microscope that allows calcium imaging in a stationary (from the microscope objective's perspective) preparation. This has been done previously, as acknowledged in the Discussion, in a paper by Migault (2018), and similar observations have been possible through an optical trapping approach established for static (Favre-Bulle 2018) and vibrational (Favre-Bulle 2020) fictive vestibular stimuli. While this new microscope design is well suited to calcium imaging in vestibular experiments on zebrafish, it is not clear that it will be taken up more broadly by other users in the broader community. The biological impact of the current manuscript is in 1) a mapping of the specificity of utricular hair cell responses to different types of vestibular stimulus, 2) a similar matching for VGNs, and 3) the organized mapping that connects these distributions between the HCs and VGNs. The first of these is not especially exciting, since it was predicted based on the anatomical orientation of the hair cells, and the last of these observations is predicted by previously-reported EM data. As such, while there are a few nice examples of novel biological insight, many of the biological results are corroborating, rather than extending, prior work.

In summary, while this represents a nice combination of innovation and targeted biological discovery, I am not convinced that this is enough for Nature Communications. It is a solid high-quality project, though, so if my opinion of impact is in the minority, I would be prepared to support publication.

Specific comments:

Comment 1:

Shouldn't the saccula also be represented in the schematic of Figure 5a?

Comment 2:

In interpreting the data in Figure 6, the point should be made more strongly just how few the roll vibration responsive neurons are (maybe 3 strongly responding neurons in the whole dataset). Given their clean responses, I am inclined to believe the result, but the scarcity of these neurons makes this less than conclusive. I would also think it worth mentioning that these are invariably very deep (ventral) in the ganglion.

Comment 3: Lines 471-473.

"Thus, the tiltable objective microscope, with exceptional flexibility in the vestibular stimulus direction/modality and the observation direction of a specimen, enables quantitative functional imaging analysis of the vestibular processing." is perhaps a little too strong, and suggests a greater leap forward in the study of vestibular processing than is warranted.

Reviewer #3 (Remarks to the Author):

This study is a beautiful tour de force to image the response of the inner ear hair cells and VGN to linear acceleration. I commend the authors for gorgeous and clear figures, as well as thorough

characterization of their system.

Overall, the manuscript is well written and the results convincing, the authors clearly state the limitations and provide adequate discussion of most issues. I have some suggestions to make the results clearer, as well as some clarifications request to the authors.

There are some overlaps with already published literature (ref 20 and 21), but the authors acknowledge them and their focus on HCs and VGN makes it different enough to be novel.

Major comments

Line 86 and 89: The 'negligible' acceleration changes are hard to assess with Sup.fig. 1 as they are too small and pixelated. Authors should report the numerical value for the maximum acceleration measured.

Sup.Fig. 2a, Fig.2d: The small negative deviations are borderline out of the noise from the ratioed fluorescence estimates. The hypothesis of hyperpolarization is sound, but could there be a registration/optical reason for their location/existence?

For the vibrational stimuli, what was the frequency? Based on the observations of frequency selectivity in larval zebrafish (especially relevant here is Favre-Bulle I, et al., Nat Com, 2020 that shaken the utricle with optical traps), could the HCs show selectivity to some frequency? This is discussed somewhat on line 499, but the authors have a great method to formally test some of these hypotheses.

Regarding the weaker response observed in the vibration paradigm, have the authors compared the HC responses to the actual acceleration measured in Sup.Fig.1 ? It seems like the acceleration experienced by the HC is weaker.

Line 393: What do the authors think the non-responsive VGN neurons could respond to? Or would it be a sensitivity issue?

Line 502: The authors mention the importance of Ca buffering in the response of HCs, GCaMP binds calcium, could this affect the properties of the HCs? Have the authors observed behavioral defects in their fish lines?

Line 552: The optical artefacts of the tiltable approach are already marked, although manageable. How would the author deal with additional motion artefacts from head movements? Furthermore, how effective would be the imaging without the removal of the ventral portions of the head described on line 635?

Minor comments

Line 57: It would be useful to say why larval zebrafish are an exceptional model.

Line 58: There are differences between mammalian and zebrafish, for example the presence of a lagenar otolith.

Line 81: 'which contained the specimen'

Line 152 'from the ventral' I assume 'side' is missing?

Fig.2 e: I am confused by the legend. Are the top graphs the average across multiple trials for one utricle and at the bottom for all 6 (or 3) utricles?

Line 318-319: articles are missing in front of utricle, sacculle, etc...

Sup.Fig.2b-c legend line 34: ' The same utricle shown in Figure 2d.' do the authors meant 2b or 2c?

Line 321: What was the time between the slices? Could the maximum intensity projection induce timing problems in the response profiles? No details are given regarding how long this volumetric acquisition was taking.

Line 453: I am unsure of why the method in 21 would not reproduce a natural otolith displacement, it cannot cover all of the possible displacements, but neither can rolls

Line 557: The fish line needs to be reported

We are grateful for the thoughtful comments and suggestions and have made substantial revisions to our manuscript by adding new experimental data, further analysis, figure panels, tables, a full description of the experimental setup, and detailed discussion. We believe that the revisions have increased the readability and impact of the manuscript and made it more accessible to the broader community.

We have addressed all reviewer comments, as described below. The reviewer comments are colored blue and italicized.

REVIEWER COMMENTS

Reviewer #1 (Remarks to the Author):

The manuscript by Tanimoto is an exemplary piece of work using a novel optical apparatus and clever stimulus paradigms to address a long-standing question in the field: how do hair cells (and sensory afferents) respond during head motion. Admittedly, given that the mechanisms of hair cell force transduction are well established, I think that there is a strong prediction for how these cells should respond. However, while the manuscript provides the expected answer, its real value is in establishing the technology to holistically and comprehensively monitor the peripheral vestibular system. The authors report intriguing topography and differential responses to stimulation at different frequencies. I have a number of suggestions for the authors that I detail below that I think would make the manuscript more accessible and/or would reach a broader audience. I also detail a number of experimental suggestions but I stress: these are entirely optional and likely beyond the scope of the paper. I trust that the editor can review the authors' responses to my suggestions and do not need to see the manuscript again — it's nearly ready for publication. I congratulate the authors on an excellent bit of work and sincerely hope that they continue to use their apparatus and acumen to address open questions in the vestibular field. I would also encourage the authors to consider posting this to bioRxiv; we have multiple manuscripts upcoming that would benefit from being able to cite this lovely work.

David Schoppik, Ph.D.

We are most grateful for your kind comments and helpful suggestions. Based on your suggestions, we have revised our manuscript as detailed below. As for posting the manuscript to a preprint platform, our manuscript has already been uploaded to Research Square (<https://doi.org/10.21203/rs.3.rs-1845426/v1>). *bioRxiv* does not recommend dual posting as it might cause confusion for readers.

Major suggestions:

1. Forces on the hair cells are a function of the motion of the utricular otolith. This new apparatus offers a unique opportunity to directly measure the motion of the otolith in response to linear acceleration. Similarly, it might be possible to measure the hair cells themselves. Such data would be of considerable interest to groups that model the utricle, such as Jong-Hoon Nam at the

University of Rochester. It might help to shed light on later questions (see below) to know if the utricle moves with the stage, and the hair cells move with the stage, or if there are non-linearities that make this challenging. If I understand the setup correctly this should be achievable by slight changes to the geometry to allow visualization of an otic capsule mounted lateral to the objective. Again, this is just a suggestion; the results of the paper are unlikely to change with these measurements.

We appreciate these valuable comments and suggestions. The movement of the otolith/hair bundles determines how hair cells respond to head motions. Measurement of the otolith/hair-bundle motion is one of our research projects. Since this is beyond the scope of the present study, we would like to prepare another manuscript in the future to address this.

*2. I found the differentiation of the two stimuli a bit confusing. If I understood the stimulus paradigm correctly, the biggest differentiator wasn't the temporal properties (i.e. low-pass vs. high-pass) but instead whether cells responded to large DC offsets and/or whether they responded to ~2Hz oscillations. One experimental suggestion would be to examine the responses (of hair cells, but perhaps more interestingly of VGNs) to the 2.2Hz stimulation *when the fish was at an eccentric angle*. We have observed [unpublished] that zebrafish can perform a vestibulo-ocular reflex in response to stimulation comparable to the 2Hz trapezoid at baseline and at more eccentric angles so presumably the nervous system can differentiate high-frequency oscillations even if they are superimposed on eccentric body postures.*

Your interpretation of the stimulus paradigm was largely correct: the biggest differentiator was whether cells responded to large DC offsets and/or whether they responded to *oscillations with multi-band frequency*; the vibration stimulus produced oscillations with multiple frequency peaks. **Supplementary Fig. 1d, g** show the acceleration time course and frequency spectra.

Based on the suggestion, we have added a description about the sustained, DC offset stimulus and the oscillatory stimulus to Line 174–177, 258–262, and 574–576. We have also added a discussion about the limitation of the vibration stimulus and future direction of the study to Line 539–553. Additionally, we have also prepared **Supplementary Fig. 1e, f** to better describe the characteristics of the stimuli used in this paper.

3. The vestibular literature is quite vast and so the authors may have chosen not to address the following areas for brevity's sake. Nonetheless, I think it would be of value to discuss the following: Ian Curthoys' work examining the responses of the utricle to high-frequency vibrations in guinea pigs, Dan Merfeld's work looking at vibration discrimination in humans, and the Beraneck/Straka work examining different frequency channels in the frog central vestibular projection neurons. I strongly encourage the editor to allow the authors sufficient space to address this recent vestibular work as I feel it would expand the impact of this manuscript.

We appreciate the suggestions and encouragement. Based on these suggestions, we have added discussion about Dr. Ian Curthoys's work to Line 669, Dr. Daniel Merfeld's work to Line 670, and Drs. Beraneck and Straka's work to Line 680.

Minor comments, line-by-line

The title is a bit challenging as it isn't clear how the paper addresses "discrimination," which usually has a specific meaning to sensory physiologists.

We have considered this and are amenable to changing the title. Changing only the word "discrimination" to another word does not make the title understandable, however. Also, the original title did not adequately represent the major findings of the study, namely, direction selectivity. Therefore, we revised the title to, "**Tilttable objective microscope visualizes selectivity for head motion direction and dynamics in zebrafish vestibular system.**"

We have also revised the last sentence in the abstract and changed the word "discrimination" to "decomposition" on Line 57, 78, 583, and 591. The phrase, "decomposition of head motion dynamics by hair cells," was used by Beraneck and Straka, *J. Vestib. Res.*, 2011. Thus, this revision ensures that the sentences are appropriate to the context.

12: "on" should be "about"

We have revised this as suggested.

16: "visualizes" should be "reveals"

We have revised this as suggested.

31: "received" should be "sensed"

We have revised this as suggested.

34: the manuscript only uses the acronym MET sparsely, consider spelling it out each time.

We have revised this as suggested.

42: Have you considered that the macula itself may not be uniform with respect to its stiffness, imparting frequency sensitivity? That may have been treated in the literature, but I am not aware of any papers in zebrafish. Also 496-501

We appreciate these insightful comments. We did not consider it in the original manuscript, but hair-bundle stiffness is one of the important, known mechanical properties that contribute to frequency sensitivity, and therefore we have added discussion of this to Line 594–598. We did not find studies that measured the stiffness of the macular epithelium other than the hair-bundle and otolithic membrane. The biophysical properties have been studied in several vertebrate species, but literature on fish is relatively sparse. Although the macula in larval zebrafish is small and likely uniform with respect to stiffness, the macula may develop regional differences with respect to stiffness as fish mature. The *in vivo* fish preparation described in this paper will be useful to address this issue.

53: The word “modality” isn’t quite correct — they are the same sensory modality (linear acceleration). This comment applies elsewhere e.g. 391

We appreciate this comment and have revised all of the phrases based on the suggestion.

61: “enable” should be “enables”

We have corrected this as suggested.

78: it would be worth noting that because the fish is off the axis of rotation, any stimulus in this apparatus will have both a linear and angular acceleration.

We have added a sentence describing this to Line 91, “**Because a specimen was off the axis of stage rotation, any stage rotation produced both linear and angular acceleration.**”

92: reference 24 would have only treated yaw head movements, but these are pitch and roll, correct?

Yes, correct. We have added clarification that reference 24 measured yaw head movements to Line 103.

Head yawing produces substantial inertial acceleration in the medial-lateral axis in the fish-centered coordinate as well as in the rotational axis in a horizontal plane. **Supplementary Figure 1d** shows that vibration stimulus produced inertial acceleration in an axis, and when a fish is placed for the roll vibration, this stimulus produces acceleration in the medial-lateral axis. Thus, it is reasonable to mention that the range of frequency components of the vibration stimulus corresponded to those of physiological head movements.

139: why is there a bigger change at 360 relative to 0? does that speak to the image processing?

This was caused by slight bleaching/photoconversion of Kaede during the imaging. The slight bleaching/photoconversion occurred largely linearly over time in green/red channels in individual cells, and was independent from the stage rotation. Therefore, the bleaching/photoconversion-induced intensity changes were removed by subtracting the linearly changing “trend lines” before calculating the green-to-red ratios. To create the trend lines, signals during periods before the initial stage rotation started and after the last stage rotation ended were separately fitted by straight lines. The trend line between these periods was estimated by a linear interpolation using the data points before the initial stage rotation started and after the last stage rotation ended. This subtraction corrected the bleaching/photoconversion-induced intensity changes for the most part in the time-series data, but because the estimation by the linear interpolation in the middle part might be slightly off the true signal changes, the deviations of the corrected signals during 360 degrees were larger than those during 0 degrees. A corresponding paragraph in the Method section was not adequate, and therefore we have revised the paragraph on Line 885–893.

This occurred only in Kaede experiments and did not occur in the beads or other fish experiments.

157: it's up to the authors, but I find "nose-up" and "nose-down" to be clearer than nose/tail.

We have considered this, but at least in the vestibular periphery, "down" has more direct biological meaning regarding the directions of stimulus, because otoliths move downward during tilts and stimulate hair cells. Therefore, we would prefer to use nose-down/tail-down/lateral-down/medial-down.

183: the arrow of the vector only points in the preferred direction, why refer to the "anti-preferred?"

Thank you for this useful comment. Description regarding the anti-preferred direction was missing in the original manuscript. We have added this to Line 207–210.

213: I'm not entirely sure that figure e adds. This is the collective response of the end organ. If we are meant to compare response magnitudes across directions, why not quantify these data and show it?

Figure 2e supplements information regarding the differences between bi-/monophasic responses: the negative responses are difficult to show in the heat-map time courses in **Fig. 2d**. Therefore, mean responses are shown by traces in **Fig. 2e**. We have added insets that show examples of the negative responses. Response magnitudes are represented by response vectors in **Fig. 2f, g**. Because of space limitations, we have added the value of response magnitudes to **Supplementary Table 1**.

*213: similarly, why are the average responses comparable given the distribution of hair cells preferred directions is not. Or are these just the average responses *of cells tuned for that direction?**

These are the average responses *of cells tuned for that direction*. The distribution of HC's preferred directions appears polarized in **Fig. 2f, g**, but because the responses were averaged across cells tuned for that direction, the responses are comparable. An anatomical study showed that there are a not-small number of morphologically developing/immature hair cells in the utricle (Liu et al., *Nat. Commun.*, 2022). To what extent these developing/immature hair cells respond is currently unknown. These cells might have averaged out the population responses.

248: I was astounded that there's really no response to the first 5 seconds of this stimulus, particularly since there is a response in the VGNs. It would be really nice to know what's going on. I appreciate the Discussion in 478-479 but it would be really helpful to propose a way to test your hypothesis.

Acceleration during the vibration stimulus (measured in **Supplementary Fig. 1d**) indicated that hair cells were stimulated intermittently. To directly examine this, hair-bundle displacement should

be imaged during the vibration stimulus. Additionally, Ca^{2+} influx through voltage-gated Ca^{2+} channels at the basolateral membrane might have caused these interesting responses. This will be examined pharmacologically. We have added a discussion of this hypothesis and experiment to Line 564–572.

263: it would be helpful to have the anatomical axes labelled in this graph (R/C and M/L)

We have added the anatomical axes to **Fig. 3c, d**.

438: do you ever see striolar/extrastriolar differentiation of axons? I ask because in 488/489 you are effectively proposing that striolar/extrastriolar are carried by different VGNs but the data don't really speak to that, do they?

In the otolith organ sensory epithelium, it appeared that the diameter of striola-innervating stem axons is larger, but because the branched peripheral axons intermingled, it was difficult to trace and clearly distinguish striola/extrastriola-innervating axons in our fluorescence microscopy data. However, reconstruction of EM serial section data from Dr. Martha Bagnall's lab showed that VGNs with myelinated axons tend to innervate striolar HCs whereas those with unmyelinated axons tend to innervate extrastriolar HCs, although there are a relatively small number of VGNs that innervate both striolar and extrastriolar HCs (Figure 5e, i in Liu et al., *Nature Communications*, 2022). Based on this EM data, vestibular signals from the striolar/extrastriolar regions are likely to be relayed through the two channels of VGNs. We have added this description to Line 581–585.

515: I'd add "inferred" before afferents. You could look at Hamling et. al., still on bioRxiv for work here from our lab in support of the qualifier.

We have added "inferred" before afferents and added Hamling et al. as a reference.

531: It's true that the vestibulospinal basal rate is low, but the rate of the inferred afferents is quite high, again, see our work on bioRxiv and Liu et. al.. So I do not think that the rate of the afferents is what is contributing the absence of negative deltaR/R0

We appreciate this valuable comment. In larval zebrafish, the firing rate of afferents is inferred based on the EPSC frequency in the vestibulospinal neurons (median frequency, ~15 Hz) in Figure 3E in Hamling et al., *bioRxiv*, 2021. The mean firing rate of afferents inferred from the EPSC frequency in the vestibulospinal neurons during low-amplitude, translational acceleration was between ~3 and ~10 Hz in Figure 6 in Liu et al., *Neuron*, 2020. Both data indicate that the inferred afferent firing rate is not markedly low in larval zebrafish, although it seems to be lower than those in mammals (mean firing rate >50 Hz, Fernandez and Goldberg, *J. Neurophysiol.*, 1976). Therefore, we have revised our manuscript (Line 646–653). Compared to the hair cells where Ca^{2+} continuously enters through large conductance channels that are open in the resting state, the intracellular Ca^{2+} level in the afferents in the resting state may not be high. Even when the afferents

decrease the firing rate during tilt, subtle changes might have been masked by the artificial fluorescent intensity changes.

540: Currently, vestibular implants target the semicircular canals because they are straightforward to map; I do not believe anyone is proposing to use topography because they aren't targeting the utricle.

We appreciate this valuable comment. We have revised accordingly.

549: this would be a perfect place to look at the Beraneck & Straka work.

We have added Beraneck & Straka's work to Line 680.

612: is the trapezoid in position? in velocity? in acceleration?

The trapezoid waveform is in position. We have added mention of this.

654: I sincerely hope you will do it with GCaMP next time!

We have! We imaged the activity of brain neurons and axial muscles with GCaMP (and tdTomato) during roll tilt using tiltable objective microscopes to investigate the biomechanics and neural circuits for posture control. We have uploaded another manuscript to *bioRxiv* (Sugioka et al., *bioRxiv*, <https://biorxiv.org/cgi/content/short/2022.09.22.508983v1>). We hope that this further work will also excite and be informative to the community who work on balance and posture control.

791: Do consider just uploading the data to Zenodo. It's free and easy to do.

We have uploaded the code and dataset to Zenodo, <https://doi.org/10.5281/zenodo.7147882>. During this process, we found a minor misrepresentation of data in **Fig. 6d** and updated the figure panel in the revised manuscript.

Reviewer #2 (Remarks to the Author):

General thoughts:

The authors design and build a spinning disc confocal microscope in which the stage can be rotated or shaken to produce vestibular stimuli during calcium imaging. Using this microscope they describe responses (to static and vibrational pitch and roll stimuli) in utricular hair cells that are expected based on previous descriptions of the cells' anatomical orientations. They also show that extrastriolar HCs preferentially report on static rotations while striolar HCs respond more strongly to vibrational stimuli.

In the vestibular ganglion (VG), a relatively small number of neurons show selective tilt

responsiveness, and a smaller number show weak roll stimuli for lateral down specifically. These are distributed in characteristic positions within the VG, suggesting spatial organization. Similar observations hold true for vibrational stimuli, although with lower numbers of responsive cells, especially for roll stimuli. Finally, they use targeted photoconversion to describe the anatomical innervation patterns that provide the spatial response patterns described for different stimuli in the VG.

The work is clear, the data (once ratiometric calculations are included) are convincing, and the interpretations are sound. Figures are very clear and generally convincing. There are some very minor grammatical issues, but the writing is generally very good.

Aside from a few minor comments included below, I believe that this work is already at publication quality, but I have concerns about the impact of the work on the field. There is technical impact in the development of a new type of microscope that allows calcium imaging in a stationary (from the microscope objective's perspective) preparation. This has been done previously, as acknowledged in the Discussion, in a paper by Migault (2018), and similar observations have been possible through an optical trapping approach established for static (Favre-Bulle 2018) and vibrational (Favre-Bulle 2020) fictive vestibular stimuli. While this new microscope design is well suited to calcium imaging in vestibular experiments on zebrafish, it is not clear that it will be taken up more broadly by other users in the broader community.

The biological impact of the current manuscript is in 1) a mapping of the specificity of utricular hair cell responses to different types of vestibular stimulus, 2) a similar matching for VGNs, and 3) the organized mapping that connects these distributions between the HCs and VGNs. The first of these is not especially exciting, since it was predicted based on the anatomical orientation of the hair cells, and the last of these observations is predicted by previously-reported EM data. As such, while there are a few nice examples of novel biological insight, many of the biological results are corroborating, rather than extending, prior work.

In summary, while this represents a nice combination of innovation and targeted biological discovery, I am not convinced that this is enough for Nature Communications. It is a solid high-quality project, though, so if my opinion of impact is in the minority, I would be prepared to support publication.

We appreciate Reviewer #2's in-depth knowledge on the vestibular research and evaluation of our work as a solid, high-quality project, and are grateful for the reviewer's flexibility regarding the decision. Because Reviewer #1 (Dr. David Schoppik) and Reviewer #3 both provided favorable comments supporting publication, we hope that Reviewer #2 is also prepared to support publication.

Regarding the technical impact of the development of a new type of microscope, Reviewer #2 stated that calcium imaging during vestibular stimulation has been done previously. However, as we described in the Discussion from Line 502–518, the previously reported methods, due to limitations in the previously reported approaches, do not necessarily produce similar observations to that which we described in this manuscript.

Migault et al. (2018) built a miniaturized rotating microscope. Although this provided physiological tilt stimulus, the feasible tilt angle range was limited within $\pm 25^\circ$. However, as we showed in **Supplementary Figs 2b-e** and **4a-d**, hair cells and VGNs exhibited maximal responses to static tilt at approximately $\pm 90^\circ$. Thus, their method could have measured responses to static tilt within a small fraction of the wide dynamic range of the vestibular system, and it was not able to examine responses to the full range of the sustained physiological vestibular inputs. Moreover, because their method enabled tilt stimulus only in the roll axis, direction selectivity in the multi-dimensional space has not been investigated.

Favre-Bulle et al. (2018; 2020) artificially displaced otoliths using an optical trapping technique. This technique can selectively stimulate specific otolith organ(s), and therefore it is useful to examine each organ's contribution to the vestibular processing. However, the displacement produced by optical trapping was small (between 15 nm and 140 nm depending on the frequency and on whether utricular or saccular otoliths are targeted). In contrast, the utricular HCs transduce hair-bundle/otolith displacement over a wide range, $\sim 1 \mu\text{m}$ to a few micrometers, in both mammals and larval zebrafish (Holt et al., *J. Neurosci.*, 1997; Tanimoto et al., *J. Neurosci.*, 2011), which suggests that the hair-bundle/otolith displacement also occurs in this wide range during physiological head motion. Thus, the optical trapping technique does not necessarily produce results that are similar to the present study. Moreover, in order to reproduce the otolith displacement *during physiological head tilts*, the optical trapping stimulus will need to be calibrated to a measured otolith displacement that occurs *during physiological head tilts*. However, spatial and temporal characteristics of the otolith displacement *during physiological head tilts* are currently unknown. Therefore, to what extent observed responses in the optical trapping papers were relevant to physiological head motion is unclear.

More critically, as we discussed on Line 515–518, the previously reported light-sheet microscopy is unlikely to enable HC and VGN imaging. This is because the location of HCs and VGNs that are ventrolateral to the brain makes it difficult to image individual cells. Additionally, the otoliths are located at the lateral side of the VGNs. The otoliths scatter excitation light sheet and prevent the imaging. Furthermore, fish preparation for imaging all of the inner ear hair cells and VGNs did not exist. Therefore, without the new, *in vivo* preparation described in the present manuscript, the previously reported methods alone are unlikely to successfully image HCs and VGNs.

We think that the existing methods are unlikely to enable observations that are similar to our results, although the previous reports greatly contributed to the visualization of brain-wide vestibular-induced activity patterns. With the new concept that only the objective lens and specimen are tilted during imaging, the tiltable objective microscope fills the gap between the existing methods and what has been sought in the vestibular research field. Thus, through publication of this manuscript, the new concept and design of this microscope can be shared to the broader community for further development of functional imaging approaches and a better understanding of the vestibular system.

As for the usability of the microscopes, we think that the tiltable objective microscope will be used more broadly by other users in the broader community than existing methods. It should be noted that the construction, optical alignment, and control of the previously reported light-sheet microscopes require specialized, highly sophisticated expertise in optics and engineering. Because this is a high threshold for biologists, these microscopes have not been used by any laboratories

other than those who originally developed the microscopes (Scott lab and Bormuth lab), to the best of our knowledge.

In contrast, building and using a tiltable objective microscope does not require special expertise. We built another tiltable objective microscope for simultaneous imaging of the activity in brain neurons/axial muscles and behavior during tilt, and by using these microscopes, we revealed the biomechanics and neuromuscular mechanisms for fine postural control behavior (Sugioka et al., *bioRxiv*, <https://biorxiv.org/cgi/content/short/2022.09.22.508983v1>). The tiltable objective microscope is composed of commercially available optomechanical components, and the confocal microscope is one of the most widely used optics for fluorescent imaging. These user-friendly features will make it easier for biologists in the broader community to examine vestibular functions.

Additionally, to increase the throughput of the tiltable objective microscope, the imaging area can be enlarged by using another objective lens and/or confocal scanner for wide-area imaging (Muto et al., *Curr. Biol.* 2013). Imaging volume can also be increased by adding an electrically driven lens positioner (Kohashi and Oda, *J. Neurosci.*, 2008). Furthermore, the spinning disk confocal optics that was used in this paper can be combined with two-photon excitation optics for *in vivo* imaging in mice (Otomo et al., *BBRC*, 2020). Thus, based on the present study, the design of the tiltable objective microscope can be used by other researchers in the broader community. We have added these discussions to Line 554–558.

Regarding the biological impact of the current manuscript, we agree that some of the biological results in this manuscript could have been predicted from previous and recent anatomical data. Reviewer #2 stated that (1) *a mapping of the specificity of utricular hair cell responses to different types of vestibular stimulus* is not especially exciting, since it was predicted based on the anatomical orientation of the hair cells. However, *anatomical orientation* of the hair cells does not predict how hair cells respond to static/dynamic vestibular stimulus. The major finding of our study, the different magnitude of responses to static/dynamic stimulus across hair cells in a sensory organ, would not be inferred from the anatomical orientation unless hair cell activity was directly quantified during tilt/vibration stimulus. Thus, although some extrapolation from anatomy might have existed, the finding of the present study has a biological impact on the field.

Reviewer #2 also stated that 3) *the organized mapping that connects distributions of spatially localized responses between the HCs and VGNs is predicted by *previously-reported* EM data*. However, the EM data had NOT been published as of the day of our manuscript submission (July 11th). Based on *Nature Communications* policy (emphasized in this Article, <https://www.nature.com/articles/s41467-020-17817-x>), any data that was still in the preprint form as of the day of manuscript submission is not regarded as *reported* data. Although we appreciate the Reviewer's in-depth knowledge of the progress in the field, the journal emphasizes the value of corroborating studies by independent groups that produce similar results through different analysis and experiments, since the corroborating work validates each other's findings and increases confidence in the scientific endeavor. Most importantly, our study and the EM paper are complementary. Although we greatly appreciate the fine anatomical EM data and noticed that this EM paper was recently published in *Nature Communications*, the anatomical data of one ear in a

fish does not fully reveal how the vestibular system works. To address this fundamental question, neural activity must be quantified during physiological vestibular stimulus in multiple ears/fish. Therefore, through publication of this manuscript, both studies will collectively and synergistically contribute to a better understanding of the vestibular system, and the readouts from the two papers will become more beneficial for the broad readership.

Specific comments:

Comment 1:

Shouldn't the saccula also be represented in the schematic of Figure 5a?

Yes, the saccule should also be represented. Thank you for noting this. Since the figure space is limited and the saccular macula is located orthogonal to the utricle, it was not easy to add the scheme of the saccular hair cells/otolith to the figure. Therefore, we have added text to the scheme to indicate that the vestibular ganglion receives inputs from the utricle, the saccule, and the semicircular canals.

Comment 2:

In interpreting the data in Figure 6, the point should be made more strongly just how few the roll vibration responsive neurons are (maybe 3 strongly responding neurons in the whole dataset). Given their clean responses, I am inclined to believe the result, but the scarcity of these neurons makes this less than conclusive. I would also think it worth mentioning that these are invariably very deep (ventral) in the ganglion.

We have revised the manuscript as suggested. The small number of vibration-responsive neurons is largely consistent with the small number of striola-innervating afferents. In the anatomical data of 104 utricular afferent neurons in larval zebrafish (Liu et al., *Nat. Commun.*, 2022), 25 neurons innervated the striola and extrastriola, and only 3 neurons exclusively innervated the striola. The relatively smaller number of roll vibration-responsive neurons than pitch vibration-responsive neurons might reflect the contribution of saccular inputs to the pitch vibration-responsive neurons. Since the saccule is supposed to respond to head motions in the pitch but not the roll axis, the pitch vibration activated more neurons than the roll vibration did. We have added this description to Line 429, 433 and 661–669.

Comment 3: Lines 471-473.

“Thus, the tiltable objective microscope, with exceptional flexibility in the vestibular stimulus direction/modality and the observation direction of a specimen, enables quantitative functional imaging analysis of the vestibular processing.” is perhaps a little too strong, and suggests a greater leap forward in the study of vestibular processing than is warranted.

We have weakened the statement in Line 535–538 in the revised manuscript as follows: **“Thus, the tiltable objective microscope, with static/dynamic vestibular stimulus in different axes and flexibility in the observation direction of a specimen, can be used for the functional imaging analysis of the vestibular processing.”**

Reviewer #3 (Remarks to the Author):

This study is a beautiful tour de force to image the response of the inner ear hair cells and VGN to linear acceleration. I commend the authors for gorgeous and clear figures, as well as thorough characterization of their system.

Overall, the manuscript is well written and the results convincing, the authors clearly state the limitations and provide adequate discussion of most issues. I have some suggestions to make the results clearer, as well as some clarifications request to the authors.

There are some overlaps with already published literature (ref 20 and 21), but the authors acknowledge them and their focus on HCs and VGN makes it different enough to be novel.

We are grateful to Reviewer #3 for the supportive comments and helpful suggestions. Based on the suggestions, we have revised our manuscript as detailed below.

Major comments

Line 86 and 89: The ‘negligible’ acceleration changes are hard to assess with Sup.fig. 1 as they are too small and pixelated. Authors should report the numerical value for the maximum acceleration measured.

We have revised the description “negligible” to “small” and added numerical values for the measured maximum acceleration to the **Supplementary Fig. 1 legend**, “90° tilt: X: 0.047 ± 0.006 g; Y: 0.159 ± 0.016 g; Z: 0.054 ± 0.006 g; 360° tilt: X: 0.030 ± 0.001 g; Y: 0.094 ± 0.003 g; Z: 0.043 ± 0.002 g; vibration: X: 0.026 ± 0.001 g; Y: 0.529 ± 0.010 g; Z: 0.031 ± 0.001 g, mean \pm SEM, 5 trials”.

The maximum acceleration during tilt was smaller than that during vibration stimulus, although there was not an order of magnitude difference between the maximum values. The small acceleration during the tilt stimulus was produced presumably because a servo-motor of the rotation stage continuously adjusted the stage to the requested angular position, and by this adjustment a small wobbling occurred. We carefully analyzed the measured acceleration during the tilt stimulus. The amplitudes of the peaks in frequency spectra were substantially smaller than those during vibration stimulus (**Supplementary Fig. 1e-g**). Because these data well represent the characteristics of the stimulus used in the study, we added these data and description to the main text on Line 95–96 and 100–105.

Regarding the magnitude of the acceleration, substantially large (>0.5 G) acceleration was produced during tilt stimulus by another standard type of rotation stage driven by a stepper motor (ThorLabs,

HDR50/M) that we tested and excluded when building the tiltable objective microscope. Thus, the rotation stage that we selected to use in this paper (ThorLabs, DDR100/M) enabled finer experiments.

Taken together, the revised manuscript with the additional data better describe the characteristics of the stimulus used in this paper.

Sup.Fig. 2a, Fig.2d: The small negative deviations are borderline out of the noise from the ratioed fluorescence estimates. The hypothesis of hyperpolarization is sound, but could there be a registration/optical reason for their location/existence?

Registration/optical reasons are unlikely to cause the localized negative deviations in the hair cell responses. We have added and analyzed experimental data, and prepared graphs in **Fig. 1h, i, n, and o**. In **Fig. 1h, n**, maximum/minimum $\Delta R/R_0$ s of fluorescent bead/Kaede-expressing neurons during tilt were plotted against the distance from the center of image rotation, which was the center of the image in **Fig. 1d, j**. The minimum $\Delta R/R_0$ did not dramatically vary with distance, although data with shorter distance tended to be slightly closer to zero. In **Fig. 1i, o**, maximum/minimum $\Delta R/R_0$ s of fluorescent beads/Kaede-expressing neurons during tilt were plotted against the angular position of the beads/neurons around the center of image rotation, which was the center of the image in **Fig. 1d, j**. The minimum $\Delta R/R_0$ did not systematically vary with the angular position, and most data were confined within $\pm 10\%$. Thus, registration/optical reasons are unlikely to cause the location of the negative deviations in the hair cell responses. These additional data better characterize the imaging system and support the hypothesis that the negative deviations reflect hyperpolarization. We have also added insets showing the negative responses to **Fig. 2e**.

For the vibrational stimuli, what was the frequency? Based on the observations of frequency selectivity in larval zebrafish (especially relevant here is Favre-Bulle I, et al., Nat Com, 2020 that shook the utricle with optical traps), could the HCs show selectivity to some frequency? This is discussed somewhat on line 499, but the authors have a great method to formally test some of these hypotheses.

The vibration stimulus contained multi-band frequency peaks of acceleration (**Supplementary Fig. 1g**). Therefore, frequency selectivity was not examined in this paper. There were three reasons for using this stimulus.

The first reason was the purpose of this paper. We tested HC/VGN responses to static/dynamic head motions. As a first step, we compared HC/VGN responses to static tilt stimulus (DC offset head motion) and vibration stimulus (oscillatory head motion). The vibration stimulus contained multi-band frequency peaks that corresponded to the frequency range of head motion during swimming. Using this vibration stimulus made it easier to compare the responses to static and dynamic head motions.

The second reason was the amplitude of the acceleration during vibration. In order to image HC/VGN responses to the vibration, a considerable level of acceleration amplitude was needed. For this reason, we purposely used angular position changes of the rotation stage in a trapezoidal

waveform (**Supplementary Fig. 1d**). This go-and-stop motion produced a relatively larger acceleration than a sinusoidal motion. In exchange for this large amplitude, ringing acceleration at the higher frequencies unavoidably occurred. Therefore, oscillation with multi-band frequency was used in this study.

The third reason was a limitation in rotation stage motion. The rotation stage does not quickly change the direction of rotation. Therefore, together with the second reason described above, it was difficult to produce acceleration changes at different frequencies. However, the tiltable objective microscope is modularly composed, and it can be combined with another stimulus device (e.g., translational platform) that can shake the otolith organs at different frequencies. Based on this paper, we would like to investigate frequency tuning of the HC/neuron responses in future studies.

We have added this limitation and future direction of study to Line 539–553.

Regarding the weaker response observed in the vibration paradigm, have the authors compared the HC responses to the actual acceleration measured in Sup.Fig.1 ? It seems like the acceleration experienced by the HC is weaker.

The maximum amplitude of the acceleration during the vibration paradigm (0.529 ± 0.010 G in the Y axis) was weaker than that during the static tilt paradigm (1 G in the Y axis). We calculated the maximum HC response amplitude per acceleration in the static tilt and vibration paradigms (maximum response $\Delta R/R_0$ [%] per acceleration [G]). The value for the vibration was smaller than that for the static tilt (tilt: 79.7 ± 1.1 %/G in pitch responses; 69.6 ± 6.1 %/G in roll responses; vibration: 24.2 ± 2.6 %/G in pitch responses; 27.0 ± 1.1 %/G in roll responses, mean \pm SEM, 6 utricles). This suggests that, even if the vibration and static tilt with the same amplitude of acceleration were applied, the hair-bundle displacement during the vibration stimulus would be smaller than that during the static tilt stimulus. We have added the analyzed data and the description to Line 271–280.

Line 393: What do the authors think the non-responsive VGN neurons could respond to? Or would it be a sensitivity issue?

As we described on Line 362–364, the rostral division of the VG contains neurons, each of which innervates the utricle, the anterior part of the saccule, or the anterior or horizontal semicircular canal. Therefore, the non-responsive VGNs could respond to linear acceleration in the dorsal-ventral axis (which the saccule receives) or angular acceleration (which the semicircular canals receive). Since the semicircular canals have not yet developed at the larval stage, a subpopulation of the non-responsive VGNs will start exhibiting responses at the later developmental stages. This is consistent with a previous study that showed that afferent neurons differentiate and form circuits before receptor cells start channeling acoustic/vestibular signals (Tanimoto et al., *J. Neurosci.*, 2009). These would probably explain the non-responsive VGNs, although a sensitivity issue was not fully excluded and reading out neural activity by Ca^{2+} imaging might have underestimated the number of responsive VGN neurons. We have added this description to Line 621–628.

Line 502: The authors mention the importance of Ca buffering in the response of HCs, GCaMP binds calcium, could this affect the properties of the HCs? Have the authors observed behavioral defects in their fish lines?

The effects of GCaMP to the properties of HCs are not fully excluded, but GCaMP-expressing fish were indistinguishable from non-expressing siblings, matured as adults, and produced healthy offspring just like wild-type fish. Therefore, we screened GCaMP-expressing fish by their fluorescence expression and used them for the experiments. Another GCaMP-expressing transgenic fish was used for the functional imaging of lateral line HCs (Jiang et al., *eLife*, 2017; Zhang et al., *Nat. Commun.*, 2018). The GCaMP-expressing fish of this transgenic line were also indistinguishable from non-expressing siblings, matured as adults, and produced healthy offspring just like wild-type fish. Therefore, it appears that GCaMP had minor effects, if any, to the properties of HCs. We have added this description to the Discussion (Line 610–613) and Methods (Line 692–697) sections.

Line 552: The optical artefacts of the tiltable approach are already marked, although manageable. How would the author deal with additional motion artefacts from head movements? Furthermore, how effective would be the imaging without the removal of the ventral portions of the head described on line 635?

Fictive fish preparation in which neuromuscular junctions are pharmacologically blocked is widely used in neurophysiological experiments. The present study used this fictive preparation. Therefore, vigorous head movements did not occur in this study.

Even in a condition in which fish head movements occur, two-color ratiometric imaging would reduce the motion-induced artefacts. This ratiometric imaging method is widely used for functional imaging in behaving animals to cancel motion-induced artefacts (Higashijima et al., *J. Neurophysiol.*, 2003; Nguyen et al., *PNAS*, 2016). As a further demonstration of the tiltable objective microscope, we successfully imaged the activity of brain neurons and contracting muscles during a roll tilt-induced behavior. We have posted another manuscript on this work (Sugioka et al., *bioRxiv*, <https://biorxiv.org/cgi/content/short/2022.09.22.508983v1>).

Imaging without the removal of the ventral portions of the head would be ideal, if it enabled imaging. However, tissues in the ventral portions of the head are not transparent, and they contain different types of cellular and non-cellular structures, which produces optical distortion. Thus, imaging without removal would currently be difficult. However, the fish preparation described in this study is new, and therefore there may be room for improvement. We will work on this in future studies.

Minor comments

Line 57: It would be useful to say why larval zebrafish are an exceptional model.

We have added the following description to Line 60: “Larval zebrafish are an exceptional model system **because the high transparency and small size of organs enable *in vivo* whole-organ imaging.**”

Line 58: There are differences between mammalian and zebrafish, for example the presence of a lagenar otolith.

We have corrected the description on Line 62: “The vestibular system is **largely** conserved among vertebrates, **although there exist some differences, such as the presence of a lagena.**”

Line 81: ‘which contained the specimen’

We have corrected this sentence on Line 88.

Line 152 ‘from the ventral’ I assume ‘side’ is missing?

Yes, we have corrected this sentence on Line 174. Thank you.

Fig.2 e: I am confused by the legend. Are the top graphs the average across multiple trials for one utricle and at the bottom for all 6 (or 3) utricules?

The top graphs **Fig. 2e** show the average response across HCs in each preference group in the utricle data shown in **Fig. 2d**. Mean time series data from 5 trials in individual HCs are averaged across HCs. For example, the number of ‘Nose-down preferred’, ‘Biphasic’ HCs is 7 in this utricle. The average and deviation across the 7 HCs are shown as the top trace in the graph.

You are correct that the bottom graphs are the average across all 6 (or 3) utricles. We have modified the legend on Line 243–245 to clarify what the individual graphs show. Related to this, we have also added a description that **Fig. 2d** is the average response of 5 trials.

Line 318-319: articles are missing in front of utricle, saccule, etc...

We have corrected this sentence on Line 363.

Sup.Fig.2b-c legend line 34: ‘The same utricle shown in Figure 2d.’ do the authors meant 2b or 2c?

Yes, we meant Fig. 2b. Thank you for pointing this out. We have corrected this on Line 42. Related to this, the utricle shown in Fig. 2c (left) is the same utricle shown in Fig. 2b. Data in Fig. 2d is also acquired from the same utricle shown in Fig. 2b. We have added this information to the legend on Line 238 and 242.

Line 321: What was the time between the slices? Could the maximum intensity projection induce

timing problems in the response profiles? No details are given regarding how long this volumetric acquisition was taking.

The time between the slices was approximately 3 min and at least 6 slices were imaged. Therefore, it took approximately 20 min to image a vestibular ganglion. Images were recorded by a series of *single-optical-slice imaging*, not by a volumetric scan; a single optical slice was imaged during a stimulus session, then the next slice was imaged during another stimulus session. Therefore, the timing of the responses to the stimulus is the same between all slices. We described this in the Methods in the original manuscript, but it might have been inadequate. Therefore, we have modified our description on Line 809. As described on Line 555 in the revised manuscript, volumetric imaging will be possible in future studies by upgrading the optics.

Line 453: I am unsure of why the method in 21 would not reproduce a natural otolith displacement, it cannot cover all of the possible displacements, but neither can rolls

The description was not adequate in the original manuscript. We focused on the otolith displacement during *physiological* head motion. To reproduce the otolith displacement that occurs during physiological head motion, the optical trapping needs to be calibrated to a measured otolith displacement during physiological head motion. However, the otolith displacement during physiological head motion is currently unknown, and therefore, to what extent observed responses induced by the artificial displacement are relevant to physiological head motion is unclear.

Optical trapping can produce many possible displacements, and a small displacement (between 15 nm and 140 nm depending on the frequency and on whether the utricular or saccular otolith is targeted) was produced in the previous study. In contrast, the utricular HCs transduce hair-bundle/otolith displacement over a wide range, ~1 μm to a few micrometers, in both mammals and larval zebrafish (Holt et al., *J. Neurosci.*, 1997; Tanimoto et al., *J. Neurosci.*, 2011), which suggests that the hair-bundle/otolith displacement also occurs in this wide range during physiological head motion. Thus, the displacement produced by optical trapping may be an order of magnitude smaller than that during physiological head tilt.

The physiological hair-bundle/otolith displacement during head motion is of great interest, and movements during static tilt in particular have not been quantified in larval fish. This will be examined using the tiltable objective microscope in future studies. Together with the existing methods, the tiltable objective microscope will synergistically contribute to a better understanding of how the vestibular system works. We have added a description of this to Line 503–510 and 569.

Line 557: The fish line needs to be reported

We have added a description of the fish lines to Line 691.

REVIEWERS' COMMENTS

Reviewer #1 (Remarks to the Author):

The revised manuscript addresses all of my comments adequately. I have no hesitations recommending it for immediate publication and congratulate the authors on a job well done.

Reviewer #2 (Remarks to the Author):

The authors have adequately addressed each of my targeted concerns. I am partially convinced by their clarifications of the study's proposed novelty and impact, and I note the other two reviewers' enthusiasm for the manuscript in this regard. Given these circumstances, I endorse the manuscript for publication in its current form.

Reviewer #3 (Remarks to the Author):

I appreciate the corrections and clarifications that the authors provided. I have no further comments and strongly support the publication of this manuscript.

Gilles Vanwalleghem, PhD